# A PAC-Bayesian Link Between Generalisation and Flat Minima

**Maxime Haddouche**                                                    MAXIME.HADDOUCHE@INRIA.FR
*Inria - Département d'Informatique de l'Ecole Normale Supérieure*
*PSL Research University*
*Paris, France*

**Paul Viallard**                                                       PAUL.VIALLARD@INRIA.FR
*Univ. Rennes, Inria, CNRS IRISA - UMR 6074*
*Rennes, France*

**Umut Şimşekli**                                                       UMUT.SIMSEKLI@INRIA.FR
*Inria - Département d'Informatique de l'Ecole Normale Supérieure*
*PSL Research University*
*Paris, France*

**Benjamin Guedj**                                                      BENJAMIN.GUEDJ@INRIA.FR
*Inria and University College London*
*London, UK*

**Editors:** Gautam Kamath and Po-Ling Loh

## Abstract

Modern machine learning usually involves predictors in the overparameterised setting (number of trained parameters greater than dataset size), and their training yields not only good performance on training data, but also good generalisation capacity. This phenomenon challenges many theoretical results, and remains an open problem. To reach a better understanding, we provide novel generalisation bounds involving gradient terms. To do so, we combine the PAC-Bayes toolbox with Poincaré and Log-Sobolev inequalities, avoiding an explicit dependency on the dimension of the predictor space. Our results highlight the positive influence of *flat minima* (being minima with a neighbourhood nearly minimising the learning problem as well) on generalisation performance, involving directly the benefits of the optimisation phase.

**Keywords:** Generalisation Bounds, PAC-Bayes, Flat Minima, Poincaré, Log-Sobolev Inequalities

## 1. Introduction

Understanding generalisation in modern machine learning problems has been a major challenge in learning theory. The goal here is to upper-bound the so-called *generalisation error*, which is the gap between the population and empirical risks, $R_{\mathcal{D}}(h) - \hat{R}_{\mathcal{S}_m}(h)$. Here, $h \in \mathbb{R}^d$ represents the parameters of a predictor, $R_{\mathcal{D}}(h) := \mathbb{E}_{\mathbf{z} \sim \mathcal{D}}[\ell(h, \mathbf{z})]$ denotes the population risk, $\mathcal{D}$ is an unknown data distribution, $\ell$ is a loss function, $\hat{R}_{\mathcal{S}_m}(h) := \frac{1}{m} \sum_{i=1}^{m} \ell(h, \mathbf{z}_i)$ is the empirical risk, and $\mathcal{S}_m := \{\mathbf{z}_1, \ldots, \mathbf{z}_m\}$ is a dataset in which each $\mathbf{z}_i$ is independent and identically distributed (*i.i.d.*) with respect to $\mathcal{D}$.

Dating back to Hochreiter and Schmidhuber (1997), it has been hypothesised that the notion of 'flatness' (or sometimes equivalently referred to as 'sharpness') is closely linked to the generalisation error: among the minima found by the learning algorithm, the flatter the minimum, the lower the generalisation error. While the initial concept of flatness was (vaguely) defined through low Kolmogorov complexity, there is no globally accepted definition of flatness. Therefore, several notions

of flatness have been considered, typically based on the second-order derivatives of the empirical risk around the local minimum found by the learning algorithm, such as $\text{trace}(\nabla_h^2 \hat{R}_{\mathcal{S}_m}(h))$ (see *e.g.* Jastrzebski et al., 2017; Wen et al., 2023).

While there have been several attempts to link some form of flatness to generalisation in a mathematically rigorous manner (Neyshabur et al., 2017; Petzka et al., 2021; Yue et al., 2023; Andriushchenko et al., 2023), mainly with the framework of 'sharpness aware minimisation' (Foret et al., 2021), it has been shown recently that flat minima do not always imply good generalisation. In fact, there are scenarios such that the flattest minima result in the worst generalisation performance compared to non-flat ones (Wen et al., 2023).

In this study, we aim to develop novel links between flatness and generalisation error from a PAC-Bayesian perspective (Shawe-Taylor and Williamson, 1997; McAllester, 1999); see also (Guedj, 2019; Hellström et al., 2023; Alquier, 2024) for an introduction. While existing work involving PAC-Bayes and flatness either uses PAC-Bayes bounds as complexity measures (Neyshabur et al., 2017; Jiang et al., 2020; Dziugaite et al., 2020; Viallard et al., 2024a) or derives bounds for specific algorithms (Negrea et al., 2019; Neu, 2021), our aim is to derive general PAC-Bayes bounds involving flatness through gradient norm. Denoting by $Q$, the probability distribution of the algorithm's output $h$, we identify sufficient conditions on $Q$ such that flatness always implies good generalisation. More precisely, we make the following contributions:

- We show that, when $Q$ satisfies the Poincaré inequality and a technical condition that we identify, we can obtain a 'time-uniform estimation' PAC-Bayes bound that mainly contains two terms:

    (i) The flatness term: either $\mathbb{E}_{\mathbf{z} \sim \mathcal{D}} \mathbb{E}_{h \sim Q}[\|\nabla_h \ell(h, \mathbf{z})\|^2]$ or $\mathbb{E}_{h \sim Q}[\frac{1}{m} \sum_{i=1}^m \|\nabla_h \ell(h, \mathbf{z}_i)\|^2]$. The latter is directly linked to the Hessian of the loss $\ell$, due to the connection between the Fisher information and the Hessian of the loss (Bickel and Doksum, 2015). For instance, under certain conditions, it can be shown that the trace is linked to gradients as $\text{trace}(\nabla_h^2 \hat{R}_{\mathcal{S}_m}(h)) = \frac{2}{m} \sum_{i=1}^m \|\nabla_h \ell(h, \mathbf{z}_i)\|^2$ (Wen et al., 2023, Lemma 4.1).

    (ii) The classical PAC-Bayesian complexity term $\text{KL}(Q, P)$, where KL denotes the Kullback-Leibler divergence and $P$ is data-independent 'prior' distribution.

- We further analyse the term $\text{KL}(Q, P)$. We show that when $Q$ is a Gibbs distribution, *i.e.* $dQ(h) \propto \exp(-\gamma \hat{R}_{\mathcal{S}_m}(h))dP(h)$ for some $\gamma > 0$ and $P$ satisfies a log-Sobolev inequality, the generalisation error can be controlled *solely* by the term: $\gamma^2 c_{LS}(P) \mathbb{E}_{h \sim Q}[\|\nabla_h \hat{R}_{\mathcal{S}_m}(h)\|^2]$, where $c_{LS}(P)$ denotes the log-Sobolev constant of the prior $P$.

- We go beyond KL divergence to link flat minima to deterministic predictors (*i.e.* when $Q$ is a Dirac distribution) through a novel Wasserstein-based generalisation bound for gradient Lipschitz loss functions.

We provide a numerical assessment of the technical condition underlying our main result, suggesting that it is suitable for neural networks on classification tasks, confirming the relevance of our bounds to better understand the generalisation ability of such models. Our results further shed light on the impact of the flatness of minima on generalisation error: when the learning algorithm ensures a sufficiently regular distribution over the parameters, the generalisation error can be directly controlled by the flatness of the region found by the algorithm.

## 2. Preliminaries

**Framework.** We consider a predictor set $\mathcal{H} \subseteq \mathbb{R}^d$ equipped with a norm $\|\cdot\|$, a data space $\mathcal{Z}$ and the space of distributions $\mathcal{M}(\mathcal{H})$ over $\mathcal{H}$. We also consider a loss function $\ell : \mathcal{H} \times \mathcal{Z} \to \mathbb{R}$. We assume that we have an *i.i.d.* dataset $\mathcal{S} = (\mathbf{z}_i)_{i \geq 1} \in \mathcal{Z}^{\mathbb{N}}$ with associated unknown data distribution $\mathcal{D}$. For each $m \geq 1$, we define $\mathcal{S}_m := \{\mathbf{z}_1, \ldots, \mathbf{z}_m\}$. In PAC-Bayes learning, we construct a data-driven posterior distribution $Q \in \mathcal{M}(\mathcal{H})$ with respect to a prior distribution $P$. To assess the generalisation ability of a predictor $h \in \mathcal{H}$, we define the *population risk* to be $R_{\mathcal{D}}(h) := \mathbb{E}_{\mathbf{z} \sim \mathcal{D}}[\ell(h, \mathbf{z})]$ and for each $m$, its empirical counterpart as $\hat{R}_{\mathcal{S}_m}(h) := \frac{1}{m} \sum_{i=1}^m \ell(h, \mathbf{z}_i)$. We also define the expected and empirical risks for $Q \in \mathcal{M}(\mathcal{H})$ as $R_{\mathcal{D}}(Q) := \mathbb{E}_{h \sim Q}[R_{\mathcal{D}}(h)]$ and $\hat{R}_{\mathcal{S}_m}(Q) := \mathbb{E}_{h \sim Q}[\hat{R}_{\mathcal{S}_m}(h)]$. PAC-Bayes bounds usually aim to control the *expected generalisation error* for each dataset size $m$, *i.e.* $R_{\mathcal{D}}(Q) - \hat{R}_{\mathcal{S}_m}(Q)$.

**Background on Poincaré and log-Sobolev inequalities.** In this work, we exploit Poincaré and log-Sobolev inequalities in the PAC-Bayes framework. We first recall their definitions: for a fixed distribution $Q \in \mathcal{M}(\mathcal{H})$, we define the *Sobolev space of order* $1$ on $\mathbb{R}^d$ as follows:

$$H^1(Q) := \left\{ f \in L^2(Q) \cap D_1(\mathbb{R}^d) \mid \|\nabla f\| \in L^2(Q) \right\},$$

where $D_1(\mathbb{R}^d)$ denotes the set of derivable functions $f : \mathbb{R}^d \to \mathbb{R}$ and $L^2(Q)$ is the space of square-integrable function. In other words, $H^1(Q)$ is the set of functions that are square-integrable, with their gradient's norm also being square-integrable.

**Definition 1 (Poincaré inequality)** *A distribution* $Q$ *satisfies a* Poincaré inequality *with constant* $c_P(Q)$ *if for all function* $f \in H^1(Q)$ *we have*

$$\operatorname*{Var}_{h \sim Q}(f(h)) \leq c_P(Q) \operatorname*{\mathbb{E}}_{h \sim Q} \left[ \|\nabla_h f(h)\|^2 \right],$$

*where* $\operatorname{Var}_{h \sim Q}(f(h)) = \mathbb{E}_{h \sim Q}[f(h) - \mathbb{E}_{h \sim Q}[f(h)]]^2$ *is the* variance *of* $f$ *with respect to* $Q$. *We then say that* $Q$ *is Poincaré with constant* $c_P(Q)$, *or that* $Q$ *is* `Poinc`$(c_P)$.

**Definition 2 (Log-Sobolev inequality)** *A distribution* $Q$ *satisfies a* log-Sobolev inequality *with constant* $c_{LS}(Q)$ *if for all function* $f \in H^1(Q)$ *we have*

$$\operatorname*{Ent}_{h \sim Q}(f^2(h)) := \operatorname*{\mathbb{E}}_{h \sim Q} \left[ f^2(h) \log \left( \frac{f^2(h)}{\mathbb{E}_{h \sim Q}[f^2(h)]} \right) \right] \leq c_{LS}(Q) \operatorname*{\mathbb{E}}_{h \sim Q} \left[ \|\nabla_h f(h)\|^2 \right],$$

*where the term* $\operatorname{Ent}_{h \sim Q}(f^2(h))$ *is the* entropy *of* $f^2$. *We then say that* $Q$ *is log-Sobolev with constant* $c_{LS}(Q)$, *or that* $Q$ *is* `L-Sob`$(c_{LS})$.

The class of Gaussian distributions is an important particular case of distributions satisfying both Poincaré and log-Sobolev inequalities; this is the subject of Proposition 3.

**Proposition 3 (Gross (1975); Brascamp and Lieb (1976); Beckner (1989))** *Given a distribution* $Q = \mathcal{N}(\mu, \Sigma)$, *where* $\mu$ *is the mean and* $\Sigma$ *is the covariance matrix in* $\mathbb{R}^d$. *Then, for any* $f \in H^1(Q)$:

$$\operatorname*{Ent}_{h \sim Q}(f^2(h)) \leq 2 \operatorname*{\mathbb{E}}_{h \sim Q} \left[ \langle \Sigma \nabla_h f(h), \nabla_h f(h) \rangle \right], \quad \text{and} \quad \operatorname*{Var}_{h \sim Q}(f(h)) \leq \operatorname*{\mathbb{E}}_{h \sim Q} \left[ \langle \Sigma \nabla_h f(h), \nabla_h f(h) \rangle \right].$$

*Thus, the distribution* $Q$ *is* `L-Sob`$(c_{LS})$ *with constant* $c_{LS}(Q) = 2\|\Sigma\|_{op}$ *and is also* `Poinc`$(c_{LS})$ *with constant* $c_{LS}(Q) = \|\Sigma\|_{op}$, *where* $\|\cdot\|_{op}$ *denotes the operator norm.*

In Proposition 3, the first inequality can be derived from the classical log-Sobolev inequality for $\mathcal{N}(\mathbf{0}, \mathrm{Id})$ stated in Gross (1975), with a change of variable. Similarly, the Poincaré inequality can be obtained through a change of variable from the Poincaré inequality for $\mathcal{N}(\mathbf{0}, \mathrm{Id})$ which is a particular case of the Brascamp-Lieb inequality for log-concave probability measures (Brascamp and Lieb, 1976) and is stated explicitly in Beckner (1989, Theorem 1).

We now focus on specific posterior distributions called *Gibbs posteriors, or Gibbs distributions*. Given a fixed loss $\ell : \mathcal{H} \times \mathcal{Z} \to \mathbb{R}$, a prior $\mathrm{P} \in \mathcal{M}(\mathcal{H})$ and a dataset $\mathcal{S}_m$, the Gibbs posterior $\mathrm{Q}_{\mathcal{S}_m}^{\gamma}$ is defined such that $d\mathrm{Q}_{\mathcal{S}_m}^{\gamma}(h) \propto \exp(-\gamma \hat{\mathrm{R}}_{\mathcal{S}_m}(h)) dP(h)$, where $\gamma > 0$ is an *inverse temperature*. Gibbs posteriors are a class of closed-form solutions for relaxation of Catoni (2007, Theorem 1.2.6) stated, for instance, in Alquier et al. (2016, Theorem 4.1). Proposition 4 shows that when the prior and the loss satisfy a few properties, then the associated Gibbs posterior is $\mathtt{L\text{-}Sob}(c_{LS})$.

**Proposition 4** *Assume that* $\mathrm{P}$ *is a probability measure on* $\mathcal{H}$ *such that* $dP(h) \propto \exp(-V(h))$ *with* $V : \mathcal{H} \to \mathbb{R}$ *a smooth function such that* $\mathrm{Hess}(V) \succeq \frac{2}{c_{LS}(\mathrm{P})}\mathrm{Id}$.[1] *Assume that* $\ell(h, \mathbf{z}) = \ell_1(h, \mathbf{z}) + \ell_2(h, \mathbf{z})$ *with* $\ell_1$ *convex, twice differentiable and* $\ell_2$ *bounded. Then for any* $\gamma > 0$*, the Gibbs posterior* $\mathrm{Q}_{\mathcal{S}_m}^{\gamma}$ *is* $\mathtt{L\text{-}Sob}(c_{LS})$ *with constant* $c_{LS}(\mathrm{Q}_{\mathcal{S}_m}^{\gamma}) = c_{LS}(\mathrm{P}) \exp\left(4\|\ell_2\|_\infty\right)$.

Proposition 4 applies, *e.g.* when $\mathrm{P}$ is a Gaussian prior $\mathrm{P} = \mathcal{N}(\mu_{\mathrm{P}}, \Sigma_{\mathrm{P}})$. Notice that in this case $c_{LS}(\mathrm{P}) = 2\|\Sigma_{\mathrm{P}}\|_{op}$. This property is a straightforward application of Chafaï (2004, Corollary 2.1) with Guionnet and Zegarlinksi (2003, Property 2.6) and is stated in Appendix A for completeness. Finally, notice that satisfying a log-Sobolev inequality is stronger than satisfying a Poincaré one. This is stated for instance in Ledoux (2006, Proposition 2.1) and properly recalled in Appendix A.

## 3. Reaching a flat minimum allows Poincaré posteriors to generalise well

In this section, we consider posterior distributions $\mathrm{Q}$ that are $\mathtt{Poinc}(c_{\mathrm{P}})$. This assumption covers the important case of Gaussian measures (Proposition 3) as well as all measures satisfying a log-Sobolev inequality (Proposition 15).

### 3.1. Time-uniform estimation PAC-Bayes bounds for heavy-tailed losses

We now focus on *time-uniform estimation* PAC-Bayes bounds, *i.e.* bounds such that there exists $\mathcal{C} \subseteq \mathcal{M}(\mathcal{H})$, and $\alpha > 0$ with probability at least $1 - \delta$, for all $\mathrm{Q} \in \mathcal{C}$, and $m > 0$, there exists $\varepsilon_m > 0$ s.t.

$$\mathrm{R}_{\mathcal{D}}(\mathrm{Q}) \leq \frac{\alpha}{m}\left[\mathrm{KL}(\mathrm{Q}, \mathrm{P}) + \log(1/\delta)\right] + \varepsilon_m.$$

Here, 'time-uniform' means that the bound holds with probability $1 - \delta$ for all $m$, and 'estimation' means that we directly control $\mathrm{R}_{\mathcal{D}}$ instead of the generalisation gap $\mathrm{R}_{\mathcal{D}} - \hat{\mathrm{R}}_{\mathcal{S}_m}$. In particular, if $\sup_{m \geq m_0} \varepsilon_m \leq \varepsilon$ for some small $\varepsilon > 0$, we interpret a time-uniform estimation bound as a *transitory fast rate*, *i.e.* a bound decaying for all $m \geq m_0$ below $2\varepsilon$ at speed $1/m$. Such a property is of interest to understand why deep neural networks rapidly acquire a good generalisation ability. Another fundamental difference between time-uniform PAC-Bayes bounds (which recently appeared in Haddouche and Guedj, 2023a; Chugg et al., 2023) is that they are linked to almost surely convergence while classical PAC-Bayes results are related to in-probability convergence. We elaborate on

---

1. The notation $A \succeq B$ means that $A - B$ is a semi-definite positive matrix.

this fundamental difference in Appendix A.3.

To obtain time-uniform estimation bounds, we exploit the notion of flat minima, *i.e.* a minimum whose neighbourhood almost minimises the loss, and this property can be attained in an overpa- rameterised setting such as neural networks once the optimisation phase has been performed. We exploit this flatness property through the gradient norm $\|\nabla_h \ell(h, \mathbf{z})\|$ of the loss *w.r.t.* the predictor $h$ for any $\mathbf{z}$. To our knowledge, this is the first attempt to do so, as Gat et al. (2022) focus on gradients with respect to the data $\nabla_{\mathbf{z}} \ell(h, \mathbf{z})$.

We first state in Assumption 5 a key assumption of our work, which intricates the data distribu- tion $\mathcal{D}$ with the posterior of interest Q.

**Assumption 5** *We then say that* $Q \in \mathcal{M}(\mathcal{H})$ *is* quadratically self-bounded *w.r.t. the loss function* $\ell : \mathcal{H} \times \mathcal{Z} \to \mathbb{R}$ *and the constant* $C > 0$ *(namely* $QSB(\ell, C)$*) if*

$$\mathbb{E}_{\mathbf{z} \sim \mathcal{D}} \left[ \left( \mathbb{E}_{h \sim Q} [\ell(h, \mathbf{z})] \right)^2 \right] \leq C R_{\mathcal{D}}(Q) = C \mathbb{E}_{\mathbf{z} \sim \mathcal{D}} \left[ \mathbb{E}_{h \sim Q} [\ell(h, \mathbf{z})] \right].$$

Assumption 5 is a relaxation of boundedness, as if $\ell : \mathcal{H} \times \mathcal{Z} \to [0, C]$ then it is $QSB(\ell, C)$. It is an alternative to the bounded expected variance assumption in anytime-valid PAC-Bayes bounds (Haddouche and Guedj, 2023a; Chugg et al., 2023). A key issue with their boundedness assumption is that it must hold for all posteriors, including those providing poor generalisation performance. Our $QSB$ assumption avoids this by intricately linking the properties of the distribution $\mathcal{D}$, the loss $\ell$ and the posterior Q. Such a design is in line with the conclusions of the recent work of Gastpar et al. (2024), inviting to derive generalisation bounds valid for specific pairs $(Q, \mathcal{D})$ (rather than uniformly valid for all such pairs) to reach sharper results. Finally, we interpret $C$ as a contraction constant that attenuates, on average, the local expansion (governed by variances of Q and $\mathcal{D}$) of the loss around the mean of Q. To illustrate the applicability of $QSB$ condition beyond bounded losses, we give a concrete example of an unbounded loss satisfying it.

**Example 1** *Assume that for any* $\mathbf{z} \in \mathcal{Z}$*, the loss* $\ell(\cdot, \mathbf{z})$ *is unbounded and L-Lipschitz, that we are in the realisable case, i.e. there exists* $h^* \in \mathcal{H}$ *such that* $\forall \mathbf{z} \in \mathcal{Z}$, $\ell(h^*, \mathbf{z}) = 0$*, and that Q is an ar- bitrary distribution with mean* $m_Q$ *and standard deviation* $\sigma_Q$ *both bounded by a certain* $K$*. Then, since we have* $\ell(h, \mathbf{z}) = \ell(h, \mathbf{z}) - \ell(h^*, \mathbf{z})$*, and by Lipschitzness and Cauchy–Schwarz's inequality, for any* $\mathbf{z} \in \mathcal{Z}$*, we can deduce that we have* $\mathbb{E}_{h \sim Q}[\ell(h, \mathbf{z})]^2 \leq \left( \mathbb{E}_{h \sim Q}[\sqrt{L\|h - h^*\|\ell(h, \mathbf{z})}] \right)^2 \leq L \mathbb{E}_{h \sim Q}[\|h - h^*\|] \mathbb{E}_{h \sim Q}[\ell(h, \mathbf{z})]$*. Finally, note that by Jensen's inequality and the bias-variance decomposition, we have* $\mathbb{E}_{h \sim Q}[\|h - h^*\|] \leq \sqrt{\mathbb{E}_{h \sim Q}[\|h - h^*\|^2]} = \sqrt{\sigma_Q^2 + \|m_Q - h^*\|^2} \leq \sqrt{K^2 + (K + \|h^*\|)^2}$*. This ensures that the QSB condition holds in this case with constant* $C = L\sqrt{K^2 + (K + \|h^*\|)^2}$*.*

We are now able to state the main result of this section.

**Theorem 6** *For any* $C > 0$*, for any* $\lambda$ *such that* $\frac{2}{C} > \lambda > 0$*, for any data-free prior* $P \in \mathcal{M}(\mathcal{H})$*, for any loss function* $\ell : \mathcal{H} \times \mathcal{Z} \to \mathbb{R}_+$*, and for any* $\delta \in (0, 1]$*, we have, with probability at least* $1 - \delta$ *over*

*the sample $\mathcal{S}$, for all $m \in \mathbb{N}^*$, for all $Q$ being $\texttt{Poinc}(c_P)$, $\texttt{QSB}(\ell, C)$, and $\ell(\cdot, \mathbf{z}) \in \mathrm{H}^1(Q)$ for all $\mathbf{z}$,*

$$
\mathrm{R}_{\mathcal{D}}(Q) \leq \frac{1}{1 - \frac{\lambda C}{2}} \left( \hat{\mathrm{R}}_{\mathcal{S}_m}(Q) + \frac{\mathrm{KL}(Q, P) + \log(1/\delta)}{\lambda m} \right)
$$
$$
+ \frac{\lambda}{2 - \lambda C} c_P(Q) \mathop{\mathbb{E}}_{\mathbf{z} \sim \mathcal{D}} \left[ \mathop{\mathbb{E}}_{h \sim Q} \left( \|\nabla_h \ell(h, \mathbf{z})\|^2 \right) \right].
$$

Theorem 6 provides a time-uniform estimation bound with $\alpha = \frac{2}{\lambda(2 - \lambda C)}$ and with the threshold $\varepsilon(Q, \mathcal{S}_m) = \frac{1}{2 - \lambda C} \left( 2\hat{\mathrm{R}}_{\mathcal{S}_m}(Q) + \lambda c_P(Q) \mathbb{E}_{\mathbf{z} \sim \mathcal{D}} [\mathbb{E}_{h \sim Q}(\|\nabla_h \ell(h, \mathbf{z})\|^2)] \right)$ for any $m \in \mathbb{N}^*$. Achieving a small $\varepsilon_m$ (and thus approaching a fast convergence rate) requires two conditions: $\hat{\mathrm{R}}_{\mathcal{S}_m}(Q) \approx 0$ and expected gradients to vanish. While the first condition is often satisfied by deep neural networks, the second holds if a flat minimum has been reached through the optimisation process. Then, setting $\lambda = 1/C$ ensures a transitory fast-rate bound of $1/m$ for any $m \in \mathbb{N}^*$. Otherwise, for a fixed $m$, setting $\lambda = m^{-\alpha}/C$ with $\alpha \in [0; 1/2]$ allows adapting the rate with respect to the behaviour of the gradients. In the case of constant gradients, we recover a convergence rate of $1/\sqrt{m}$, at the cost of the time-uniform property, matching Alquier et al. (2016, Theorem 4.1).

**On the role of flat minima in PAC-Bayes learning.** We highlight that the gradient term in Theorem 6 is derived with respect to a predictor $h \in \mathcal{H}$ and not $\mathbf{z} \in \mathcal{Z}$ which is, to our knowledge, novel in PAC-Bayes. This is particularly impacting, as $\nabla_h \ell(h, \mathbf{z})$ is the gradient involved in learning procedures and, when averaged over $Q$, provides information about the nature of the minima reached and its neighbourhood; we elaborate further in Appendix A.4. Theorem 6 suggests that, to attain good generalisation ability, the mean of $Q$ must be close to two minima: *(i)* on $\hat{\mathrm{R}}_{\mathcal{S}_m}(Q)$ in order to make $\hat{\mathrm{R}}_{\mathcal{S}_m}(Q)$ small, and *(ii)* on $\mathbb{E}_{\mathbf{z} \sim \mathcal{D}}[\|\nabla_h \ell(h, \mathbf{z})\|^2]$ to ensure the gradients are small. The variance of $Q$ must fit the flatness of these minima to reduce the expected terms on the right-hand side of Theorem 6. Finally, the KL term invites, for Gaussian distributions, considering high variances and flat minima to maintain a small value for the bound.

**A focus on $C$.** Taking $\lambda = 1/C$ in Theorem 6 reduces the influence of the prior distribution P while amplifying the gradient term. Therefore, a small $C$ is desirable when working with flat minima to mitigate the effects of a poorly chosen prior. Having a small $C$ is reachable in practice: we show in Section 6, for a classification task on MNIST, that the $\texttt{QSB}$ assumption holds with $C$ strictly smaller than 1 when considering neural networks.

**Proof of Theorem 6** We start from Chugg et al. (2023, Corollary 17) instantiated with a single $\lambda$, an *i.i.d.* dataset and a prior P. With probability at least $1 - \delta$, for all $Q \in \mathcal{M}(\mathcal{H})$ and $m \in \mathbb{N}^*$, we have

$$
\mathrm{R}_{\mathcal{D}}(Q) \leq \hat{\mathrm{R}}_{\mathcal{S}_m}(Q) + \frac{\mathrm{KL}(Q, P) + \log(1/\delta)}{\lambda m} + \frac{\lambda}{2} \left( \mathop{\mathbb{E}}_{h \sim Q} \left[ \mathop{\mathbb{E}}_{\mathbf{z} \sim \mathcal{D}} [\ell(h, \mathbf{z})^2] \right] \right),
$$

where $\mathbf{z} \sim \mathcal{D}$ is independent from $\mathcal{S}$. We study the term $\mathbb{E}_{h \sim Q}[\mathbb{E}_{\mathbf{z} \sim \mathcal{D}}[\ell(h, \mathbf{z})^2]]$ on the right-hand side. We first apply Fubini's theorem to obtain

$$
\mathop{\mathbb{E}}_{h \sim Q} \left[ \mathop{\mathbb{E}}_{\mathbf{z} \sim \mathcal{D}} [\ell(h, \mathbf{z})^2] \right] = \mathop{\mathbb{E}}_{\mathbf{z} \sim \mathcal{D}} \left[ \mathop{\mathbb{E}}_{h \sim Q} [\ell(h, \mathbf{z})^2] \right] = \mathop{\mathbb{E}}_{\mathbf{z} \sim \mathcal{D}} \left[ \mathop{\mathrm{Var}}_{h \sim Q} (\ell(h, \mathbf{z})) + \left( \mathop{\mathbb{E}}_{h \sim Q} [\ell(h, \mathbf{z})] \right)^2 \right].
$$

As for any $\mathbf{z} \in \mathcal{Z}$, we have $\ell(\cdot, \mathbf{z}) \in \mathrm{H}^1(\mathrm{Q})$, we apply Poincaré inequality to obtain

$$
\mathop{\mathbb{E}}_{h \sim \mathrm{Q}} \left[ \mathop{\mathbb{E}}_{\mathbf{z} \sim \mathcal{D}} [\ell(h, \mathbf{z})^2] \right] \leq \mathop{\mathbb{E}}_{\mathbf{z} \sim \mathcal{D}} \left[ c_P(\mathrm{Q}) \mathop{\mathbb{E}}_{h \sim \mathrm{Q}} \left( \|\nabla_h \ell(h, \mathbf{z})\|^2 \right) + \left( \mathop{\mathbb{E}}_{h \sim \mathrm{Q}} [\ell(h, \mathbf{z})] \right)^2 \right].
$$

Using that Q is $\mathrm{QSB}(\ell, C)$ and re-organising the terms gives

$$
\mathrm{R}_{\mathcal{D}}(\mathrm{Q}) \leq \frac{1}{1 - \frac{\lambda C}{2}} \left( \hat{\mathrm{R}}_{\mathcal{S}_m}(\mathrm{Q}) + \frac{\mathrm{KL}(\mathrm{Q}, \mathrm{P}) + \log(1/\delta)}{\lambda m} \right)
$$
$$
+ \frac{\lambda}{2 - \lambda C} c_P(\mathrm{Q}) \mathop{\mathbb{E}}_{\mathbf{z} \sim \mathcal{D}} \left[ \mathop{\mathbb{E}}_{h \sim \mathrm{Q}} \left( \|\nabla_h \ell(h, \mathbf{z})\|^2 \right) \right].
$$

$\blacksquare$

When the $\mathrm{QSB}$ assumption is not verified, it is still possible to exploit the benefit of flat minima in PAC-Bayes at the cost of an upper bound on $\mathrm{R}_{\mathcal{D}}(\mathrm{Q})$ and a supplementary Poincaré assumption on the data distribution $\mathcal{D}$.

**Corollary 7** *For any $C>0$, for any $\lambda$ such that $\frac{2}{C}>\lambda>0$, for any data-free prior $\mathrm{P} \in \mathcal{M}(\mathcal{H})$, for any loss function $\ell : \mathcal{H} \times \mathcal{Z} \to \mathbb{R}_+$ such that $\ell(h, \cdot)$ is $\mathcal{C}^1$ almost everywhere on $\mathcal{Z}$, any $\delta \in (0,1]$, if the data distribution $\mathcal{D}$ is $\mathrm{Poinc}(c_\mathrm{P})$, with probability at least $1 - \delta$ over the sample $\mathcal{S}$, for any $m \in \mathbb{N}^*$, any posterior $\mathrm{Q}$ being $\mathrm{Poinc}(c_\mathrm{P})$ with $\mathrm{R}_{\mathcal{D}}(\mathrm{Q}) \leq C$ and such that for any $\mathbf{z} \in \mathcal{Z}$, $\ell(\cdot, \mathbf{z}) \in \mathrm{H}^1(\mathrm{Q})$:*

$$
\mathrm{R}_{\mathcal{D}}(\mathrm{Q}) \leq \frac{1}{1 - \frac{\lambda C}{2}} \left( \hat{\mathrm{R}}_{\mathcal{S}_m}(\mathrm{Q}) + \frac{\mathrm{KL}(\mathrm{Q}, \mathrm{P}) + \log(1/\delta)}{\lambda m} \right)
$$
$$
+ \frac{\lambda}{2 - \lambda C} \left( c_P(\mathrm{Q}) \mathop{\mathbb{E}}_{\mathbf{z} \sim \mathcal{D}} \left[ \mathop{\mathbb{E}}_{h \sim \mathrm{Q}} \left( \|\nabla_h \ell(h, \mathbf{z})\|^2 \right) \right] + c_P(\mathcal{D}) \mathop{\mathbb{E}}_{\mathbf{z} \sim \mathcal{D}} \left( \left\| \mathop{\mathbb{E}}_{h \sim \mathrm{Q}} [\nabla_{\mathbf{z}} \ell(h, \mathbf{z})] \right\|^2 \right) \right).
$$

The proof is deferred to Appendix C.1. Corollary 7 states that if Q reaches a flat minimum (meaning $\|\nabla_h \ell(h, \mathbf{z})\|$ is small) and this minimum is robust to the training dataset (meaning $\|\nabla_{\mathbf{z}} \ell(h, \mathbf{z})\|$ is small), then a time-uniform estimation bound is attainable with small $\varepsilon_m$, requiring only an upper bound on $\mathrm{R}_{\mathcal{D}}(\mathrm{Q})$. The assumption of small $\|\nabla_{\mathbf{z}} \ell(h, \mathbf{z})\|$ can be attained with algorithms such as Sharpness-Aware Minimisation (Foret et al., 2021). Another lead would be to focus on specific predictors where this gradient is directly controlled, as in Lipschitz neural networks training or in adversarial robustness training (Madry et al., 2018; Li et al., 2019).

Corollary 7 holds when $\mathcal{D}$ is $\mathrm{Poinc}(c_\mathrm{P})$, encompassing the case of Gaussian mixtures (Schlichting, 2019), which can approximate any smooth density (as recalled in Gat et al., 2022). However, the Poincaré constant of a general mixture is unknown, and the upper bound of Schlichting (2019) scales with the number of components, involving potentially high $\chi^2$ divergences.

**Comparison with Gat et al. (2022).** We compare Corollary 7 with Gat et al. (2022, Theorems 3.5 and 3.6). First, our result holds under the assumption that the distribution $\mathcal{D}$ follows a Poincaré inequality, which is strictly less restrictive than assuming a log-Sobolev inequality (Proposition 15). Second, they assume a bounded loss and focus solely on classification tasks satisfying a technical

assumption (see their Lemma 3.3) while ours holds for any learning problem at the sole assumption of a bounded $R_{\mathcal{D}}(Q)$, allowing the loss $\ell$ to be unbounded (and non-negative). To conclude their proof, Gat et al. (2022) use a uniform bound on $\mathbb{E}_{\mathbf{z} \sim \mathcal{D}}[\|\nabla_{\mathbf{z}} \ell(h, \mathbf{z})\|]$ in their Theorem 3.5 to have a tractable bound, diminishing the benefits of gradient norm. While they address this limitation in Gat et al. (2022, Theorem 3.6), the explicit influence of the gradient norm appears within an exponential moment on the losses (attenuated by a logarithm), averaged *w.r.t.* the data-free prior P. Thus, the associated gradients have no apparent reason to be small, and their result cannot be linked to flat minima. In contrast, Corollary 7 involves expected gradients *w.r.t.* the posterior Q, which may reach flat minima.

### 3.2. Towards fully empirical bound for gradient-Lipschitz functions

In this section, we assume that the gradient $\nabla_h \ell(h, \mathbf{z})$ is $G$-Lipschitz for any $\mathbf{z} \in \mathcal{Z}$, which is a classical assumption in optimisation, especially when considering non-convex objectives for SGD (Ghadimi and Lan, 2013; Panageas and Piliouras, 2017; Garrigos and Gower, 2023). A large portion of high-probability PAC-Bayes bounds are fully empirical, meaning that the right-hand side of the bounds can be computed. This has numerous advantages, including in-training numerical evaluation of the bound and the development of novel PAC-Bayesian algorithms that minimises such empirical bounds; see (Dziugaite and Roy, 2017; Pérez-Ortiz et al., 2021; Viallard et al., 2023b) among others. However, Theorem 6 and Corollary 7 are not fully empirical, as they involve terms such as $\mathbb{E}_{\mathbf{z} \sim \mathcal{D}}[\mathbb{E}_{h \sim Q}[\|\nabla_h \ell(h, \mathbf{z})\|^2]]$ and $\mathbb{E}_{\mathbf{z} \sim \mathcal{D}}[\|\mathbb{E}_{h \sim Q}[\nabla_{\mathbf{z}} \ell(h, \mathbf{z})]\|^2]$, which involves an expectation over $\mathbf{z} \sim \mathcal{D}$ and are thus not computable in practice. As a result, they lack the desirable properties of fully empirical bounds; we address this issue in Theorem 8.

**Theorem 8** *For any $C_1, C_2, c > 0$, for any data-free prior $P \in \mathcal{M}(\mathcal{H})$, for any loss function $\ell : \mathcal{H} \times \mathcal{Z} \to \mathbb{R}_+$ being $\mathcal{C}^2$ and for any $\delta \in (0, 1]$, we have, with probability at least $1 - \delta$ over the sample $\mathcal{S}$, for all $m \in \mathbb{N}^*$, for all $Q$ being $Poinc(c_P){=}c$, $QSB(\ell, C_1)$, $QSB\left(\|\nabla_h \ell\|^2, C_2\right)$, and $\ell(\cdot, \mathbf{z}) \in H^1(Q)$, and $\|\nabla_h \ell(\cdot, \mathbf{z})\|^2 \in H^1(Q)$ for all $\mathbf{z}$,*

$$
R_{\mathcal{D}}(Q) \leq 2\hat{R}_{\mathcal{S}_m}(Q) + \frac{2c}{C_1} \mathop{\mathbb{E}}_{h \sim Q}\left[\frac{1}{m}\sum_{i=1}^{m}\|\nabla_h \ell(h, \mathbf{z}_i)\|^2\right]
$$
$$
+ 2\left(C_1 + c\frac{4cG^2 + C_2}{C_1}\right)\frac{\mathrm{KL}(Q, P) + \log(2/\delta)}{m}.
$$

The proof is deferred to Appendix C.2. We showed that to attain fast rates, the $QSB$ assumption must hold for both the loss and its gradient. We are then able to derive an empirical generalisation bound, involving both empirical loss and gradients.

As Theorem 8 is fully empirical, it can be transformed into a generalisation metric, *i.e.* an empirical function of the predictor whose increase or decrease is correlated to the increase or decrease of the generalisation ability of the predictor. In the case of PAC-Bayes, the generalisation metric comes from the generalisation bound. Such an idea has been exploited recently (Neyshabur et al., 2017; Jiang et al., 2020; Dziugaite et al., 2020; Viallard et al., 2024a) to show that flatness of the empirical risk was correlated to generalisation. In particular, from $\hat{R}_{\mathcal{S}}(Q)$, Neyshabur et al. (2017) derived a notion of *sharpness*, stated in Equation (1), which gives information about the flatness of

the reached minima for any $Q = \mathcal{N}(\mu_Q, \sigma^2 \mathrm{Id})$. This notion is defined by

$$\underset{\nu \sim \mathcal{N}(\mathbf{0}, \sigma^2 \mathrm{Id})}{\mathbb{E}} \left[ \hat{\mathrm{R}}_{\mathcal{S}_m}(\mu_Q + \nu) - \hat{\mathrm{R}}_{\mathcal{S}_m}(\mu_Q) \right]. \tag{1}$$

Sharpness is then the averaged risk of a predictor drawn under $\mathcal{N}(\mu_Q, \sigma^2 \mathrm{Id})$ *w.r.t.* its mean Q. Theorem 8 enhance this notion of sharpness with empirical gradients when Q is $\mathrm{QSB}(\ell, C_1)$:

$$\mathrm{Sharp}_{\frac{\sigma^2}{C_1}}(Q) := \underset{\nu \sim \mathcal{N}(\mathbf{0}, \sigma^2 \mathrm{Id})}{\mathbb{E}} \left[ 2 \Big[ \hat{\mathrm{R}}_{\mathcal{S}_m}(\mu_Q + \nu) - \hat{\mathrm{R}}_{\mathcal{S}_m}(\mu_Q) \Big] + \frac{\sigma^2}{C_1} \Big[ \hat{\mathrm{G}}_{\mathcal{S}_m}(\mu_Q + \nu) - \hat{\mathrm{G}}_{\mathcal{S}_m}(\mu_Q) \Big] \right], \tag{2}$$

where $\hat{\mathrm{G}}_{\mathcal{S}_m}(h) = \frac{1}{m} \sum_{i=1}^m \|\nabla_h \ell(h, \mathbf{z}_i)\|^2$. This gradient term can be seen as the norm of an empirical Fisher information, linked to the second-order moment derivative. Thus, Equation (2) involves a notion of flatness on both the loss and its gradient, unlike Equation (1). For the sake of clarity, we specialise Theorem 8 in Corollary 9 to Gaussian distributions, introducing this notion of sharpness.

**Corollary 9** *For any $C_1, C_2 > 0$, for any data-free prior $\mathrm{P} = \mathcal{N}(\mu_P, \sigma^2 \mathrm{Id})$ with fixed variance $\sigma^2 > 0$, for any loss function $\ell : \mathcal{H} \times \mathcal{Z} \to \mathbb{R}_+$ being $\mathcal{C}^2$ for and any $\delta \in (0, 1]$, we have, with probability at least $1 - \delta$ over the sample $\mathcal{S}$, for all $m \in \mathbb{N}^*$, for all $Q = \mathcal{N}(\mu_Q, \sigma^2 \mathrm{Id})$ being $\mathrm{QSB}(\ell, C_1)$, $\mathrm{QSB}\left(\|\nabla_h \ell\|^2, C_2\right)$ and $\ell(\cdot, \mathbf{z}) \in \mathrm{H}^1(Q)$, and $\|\nabla_h \ell(\cdot, \mathbf{z})\|^2 \in \mathrm{H}^1(Q)$ for all $\mathbf{z}$,*

$$\mathrm{R}_{\mathcal{D}}(Q) \leq 2\hat{\mathrm{R}}_{\mathcal{S}_m}(\mu_Q) + \hat{G}_{\mathcal{S}_m}(\mu_Q) + \mathrm{Sharp}_{\frac{\sigma^2}{C_1}}(Q) + \mathcal{O}\left(\frac{\mathrm{KL}(Q, \mathrm{P}) + \log(2/\delta)}{m}\right).$$

## 4. Generalisation ability of Gibbs distributions with a log-Sobolev prior

A limitation of the results in Section 3 is that the KL divergence term remains generally uncontrolled, as its formulation depends on the nature of P and Q. While a closed form exists for Gaussian distributions, a natural question is whether it is possible to explicitly control the KL term for another class of distributions. Following the approach of Catoni (2007), we focus in this section on Gibbs posteriors, which naturally arise in PAC-Bayes through the use of tools from statistical physics. We show that log-Sobolev inequalities allow us to control the KL divergence of such distributions with respect to their priors.

**Controlling the KL divergence when Q is a Gibbs posterior.** Lemma 10 exploits the fact that the KL divergence can be expressed as an entropy with respect to the prior distribution P. It shows that the KL divergence of the Gibbs posterior $Q_{\mathcal{S}_m}^\gamma$ is upper-bounded by gradient terms, provided that the prior P satisfies a log-Sobolev inequality.

**Lemma 10** *For any $\gamma > 0$, for any $m \in \mathbb{N}^*$, for any data-free prior $\mathrm{P} \in \mathcal{M}(\mathcal{H})$ being $\mathrm{L\text{-}Sob}(c_{LS})$, for any loss function $\ell : \mathcal{H} \times \mathcal{Z} \to \mathbb{R}_+$ such that $\ell(\cdot, \mathbf{z}) \in \mathrm{H}^1(\mathrm{P})$ for any $\mathbf{z}$, we have*

$$\mathrm{KL}\left(Q_{\mathcal{S}_m}^\gamma, \mathrm{P}\right) \leq \frac{\gamma^2 c_{LS}(\mathrm{P})}{4} \underset{h \sim Q_{\mathcal{S}_m}^\gamma}{\mathbb{E}} \left[ \|\nabla_h \hat{\mathrm{R}}_{\mathcal{S}_m}(h)\|^2 \right].$$

The proof is deferred to Appendix C.3. The key message of this lemma is that for Gibbs posteriors, the expansion of the KL divergence is controlled by an expected empirical gradient term. Note in this case that, while the KL divergence has an explicit formulation, it requires calculating the exponential moment $\mathbb{E}_{h \sim \mathrm{P}}[\exp(-\gamma \hat{\mathrm{R}}_{\mathcal{S}_m}(h))]$ which is costly in practice. In contrast, we only need to estimate a second-order moment over $Q_{\mathcal{S}_m}^\gamma$.

**Generalisation ability of Gibbs posteriors.** When Gibbs posteriors are involved, the KL divergence can be controlled by a gradient term. An ideal approach, as in Section 3, would be to involve a Poincaré inequality. However, Gibbs posteriors do not necessarily satisfy a Poincaré inequality as in Section 3, so we need to make supplementary assumptions about the loss.

**Theorem 11** *We first have $C > 0$, $\gamma > 0$, and a data-free prior $\mathrm{P} \in \mathcal{M}(\mathcal{H})$ being $\mathtt{L-Sob}(c_{LS})$.*
(i) *For any loss function $\ell : \mathcal{H} \times \mathcal{Z} \to [0,1]$ and for any $\delta \in (0,1]$, with probability at least $1 - \delta$ over the sample $\mathcal{S}$, for any $m \in \mathbb{N}^*$, we have*

$$\mathrm{R}_{\mathcal{D}}(\mathrm{Q}_{\mathcal{S}_m}^{\gamma}) \leq 2 \left( \hat{\mathrm{R}}_{\mathcal{S}_m}(\mathrm{Q}_{\mathcal{S}_m}^{\gamma}) + \frac{\gamma^2 c_{LS}(\mathrm{P})}{4m} \underset{h \sim \mathrm{Q}_{\mathcal{S}_m}^{\gamma}}{\mathbb{E}} \left[ \|\nabla_h \hat{\mathrm{R}}_{\mathcal{S}_m}(h)\|^2 \right] + \frac{\log(1/\delta)}{m} \right).$$

(ii) *For any loss function $\ell$ and prior $\mathrm{P}$ satisfying the conditions of Proposition 4. Then, for any $\frac{2}{C} > \lambda > 0$, with probability at least $1 - \delta$ over the sample $\mathcal{S}$, for all $m \in \mathbb{N}^*$, assuming $\ell(\cdot, \mathbf{z}) \in \mathrm{H}^1(\mathrm{Q}_{\mathcal{S}_m}^{\gamma})$ and that $\mathrm{Q}_{\mathcal{S}_m}^{\gamma}$ is $\mathtt{QSB}(C)$, we have*

$$\mathrm{R}_{\mathcal{D}}(\mathrm{Q}_{\mathcal{S}_m}^{\gamma}) \leq \frac{1}{1 - \frac{\lambda C}{2}} \left( \hat{\mathrm{R}}_{\mathcal{S}_m}(\mathrm{Q}_{\mathcal{S}_m}^{\gamma}) + \frac{\gamma^2 c_{LS}(\mathrm{P})}{4\lambda m} \underset{h \sim \mathrm{Q}_{\mathcal{S}_m}^{\gamma}}{\mathbb{E}} \left[ \|\nabla_h \hat{\mathrm{R}}_{\mathcal{S}_m}(h)\|^2 \right] + \frac{\log(1/\delta)}{\lambda m} \right)$$
$$+ \frac{\lambda e^{4\|\ell_2\|_{\infty}} c_{LS}(\mathrm{P})}{4 - 2\lambda C} \underset{\mathbf{z} \sim \mathcal{D}}{\mathbb{E}} \left[ \underset{h \sim \mathrm{Q}_{\mathcal{S}_m}^{\gamma}}{\mathbb{E}} \left( \|\nabla_h \ell(h, \mathbf{z})\|^2 \right) \right].$$

The proof is deferred to Appendix C.4. Note that we could have also derived a result analogous to Corollary 7 at the cost of an additional Poincaré assumption on $\mathcal{D}$. The influence of the inverse temperature $\gamma$ is quadratic: this is the price to pay to fit the dataset and reduce the influence of the prior. This dependency is attenuated by a gradient term, which is small if a flat minimum on the $\hat{\mathrm{R}}_{\mathcal{S}_m}(h)$ has been reached. This suggests that in the case of Gibbs posteriors with a log-Sobolev prior, reaching a flat minimum on $\hat{\mathrm{R}}_{\mathcal{S}_m}(h)$ controls not only $\hat{\mathrm{R}}_{\mathcal{S}_m}(\mathrm{Q})$, but also the KL divergence and this last property is not reachable when considering Poincaré distributions. The other gradient term comes from Section 3 and requires to be close to a flat minimum on $\mathrm{R}_{\mathcal{D}}(h)$ to attain fast rates.

**Comparison to literature.** To our knowledge, existing bounds for Gibbs posteriors do not involve the gradient norm of the posterior. For instance, Zhang (2006, Theorem 4.2) involves an exponential moment with respect to the prior distribution, while Kuzborskij et al. (2019, Equation 8) consider flatness through ellipsoids around a minimum, controlling it via an 'effective dimension', which is a function of the eigenvalue of the Hessian of the theoretical risk. Finally, Rivasplata et al. (2020, Equation 6) propose a bound for bounded losses, removing the KL divergence when Gibbs posteriors are considered, with a rate of $\mathcal{O}(1/\sqrt{m} + \gamma/m)$. Assuming the Gibbs posterior reaches a flat minimum such that $\gamma \mathbb{E}_{h \sim \mathrm{Q}_{\mathcal{S}_m}^{\gamma}} [\|\nabla_h \hat{\mathrm{R}}_{\mathcal{S}_m}(h)\|^2] \leq 1$, then Theorem 11 yields a bound of magnitude $\mathcal{O}(\hat{\mathrm{R}}_{\mathcal{S}_m}(\mathrm{Q}_{\mathcal{S}_m}^{\gamma}) + 1/m + \gamma/m)$, representing a transitory fast rate with threshold $\hat{\mathrm{R}}_{\mathcal{S}_m}(\mathrm{Q}_{\mathcal{S}_m}^{\gamma})$.

## 5. On the benefits of the gradient norm in Wasserstein PAC-Bayes learning

In Sections 3 and 4, we provided various generalisation bounds, benefiting from flat minima. However, our results involve a KL divergence, implying absolute continuity of Q with respect to P, making them incompatible with deterministic predictors (obtained with Dirac distributions). Then

the following question arises: can we benefit from the desirable properties of flat minima in a PAC-Bayes bound valid for deterministic predictors? To address this issue, a recent line of work has emerged, focusing on integral probability metrics, with particular attention to the 1-Wasserstein distance (Amit et al., 2022; Haddouche and Guedj, 2023b; Viallard et al., 2023b, 2024b); see Definition 16. The idea behind these works is to replace the change of measure inequality (Csiszár, 1975; Donsker and Varadhan, 1976) with the Kantorovich-Rubinstein duality (Villani, 2009), trading a KL divergence for a Wasserstein distance. This not only yields sound theoretical bounds but also PAC-Bayes algorithms for deterministic predictors and Lipschitz neural networks (Viallard et al., 2023b). We go further here by obtaining the first PAC-Bayesian bound directly involving a 2-Wasserstein distance, trading a Lipschitz assumption for a gradient-Lipschitz one, which is well-suited for optimisation. To do so, we derive a novel change of measure inequality tailored for the condition $(\star)$ described below, which is a relaxation of the gradient Lipschitz assumption.

$$f : \mathcal{H} \to \mathbb{R} \text{ satisfies } (\star) \Leftrightarrow \exists G > 0, \forall (a, a') \in \mathcal{H}^2, \langle \nabla f(\cdot), a - a' \rangle \text{ is } G\|a - a'\|\text{-Lipschitz.}$$

**Theorem 12** *For any predictor set $\mathcal{H}$ with finite diameter $D > 0$, and for any function $f : \mathcal{H} \to \mathbb{R}$ satisfying $(\star)$, we have for all distributions $\mathrm{P} \in \mathcal{M}(\mathcal{H})$ and $\mathrm{Q} \in \mathcal{M}(\mathcal{H})$*

$$\underset{h \sim \mathrm{Q}}{\mathbb{E}}[f(h)] \leq \frac{G}{2} W_2^2(\mathrm{Q}, \mathrm{P}) + \underset{h \sim \mathrm{P}}{\mathbb{E}}[f(h)] + D \underset{h \sim \mathrm{Q}}{\mathbb{E}}[\|\nabla f(h)\|].$$

The proof is deferred to Appendix C.5. Theorem 12 shows that, when gradients are Lipschitz, it is possible to obtain a duality formula involving the gradient of the considered function, at the cost of a linear dependency on the diameter $D$ of $\mathcal{H}$. Theorem 12 is also linked to the change of measure inequality (Csiszár, 1975; Donsker and Varadhan, 1976) when the prior distribution satisfies a log-Sobolev inequality. This link is detailed in Corollary 13.

**Corollary 13** *For any distribution $\mathrm{P}$ being $\mathtt{L-Sob}(c_{LS})$ such that $d\mathrm{P}(h) \propto \exp(-V(h))dh$, with $V$ being $\mathcal{C}^2$, for any $R > 0$, for any function $f$ on the centred ball $\mathcal{B}(\mathbf{0}, R)$ of radius $R$ satisfying the $(\star)$ assumption, and for any distribution $\mathrm{Q} \in \mathcal{M}(\mathcal{H})$, we have:*

$$\underset{h \sim \mathrm{Q}}{\mathbb{E}}[f(\mathcal{P}_R(h))] \leq \frac{G c_{LS}(\mathrm{P})}{4} \mathrm{KL}(\mathrm{Q}, \mathrm{P}) + \underset{h \sim \mathrm{P}}{\mathbb{E}}[f(\mathcal{P}_R(h))] + 2R \underset{h \sim \mathrm{Q}}{\mathbb{E}}[\|\nabla_h f(\mathcal{P}_R(h))\|],$$

*where $\mathcal{P}_R$ denotes the Euclidean projection onto $\mathcal{B}(\mathbf{0}, R)$.*

The proof is deferred to Appendix C.6. Corollary 13 involves a KL divergence and an Euclidean predictor space $\mathcal{H} = \mathbb{R}^d$. This comes at the cost of approximating $\mathrm{Q}$ and $\mathrm{P}$ by, respectively, $\mathcal{P}_R \# \mathrm{Q}$ and $\mathcal{P}_R \# \mathrm{P}$. Thus, the radius $R$ is now a hyperparameter balancing the tradeoff between the quality of our approximations and the looseness of the bound (if the gradient norm is large). A notable strength is that the smoothness assumption is relaxed to apply only within the centred ball $\mathcal{B}(\mathbf{0}, R)$.

From Theorem 12, we now derive a novel generalisation bound allowing deterministic predictors.

**Theorem 14** *Let $\delta \in (0, 1)$ and $P \in \mathcal{M}(\mathcal{H})$ a data-free prior. Assume $\mathcal{H}$ has a finite diameter $D > 0$, for any loss function $\ell : \mathcal{H} \times \mathcal{Z} \to \mathbb{R}_+$ and any $m \in \mathbb{N}^*$, the generalisation gap $h \mapsto$*

$R_{\mathcal{D}}(h) - \hat{R}_{\mathcal{S}_m}(h)$ *satisfies* $(\star)$. *Assume that* $\mathbb{E}_{h \sim P} \mathbb{E}_{\mathbf{z} \sim \mathcal{D}}[\ell(h, \mathbf{z})^2] \leq \sigma^2$, *then the following holds with probability at least* $1 - \delta$, *for any* $m > 0$ *and any* $Q \in \mathcal{M}(\mathcal{H})$:

$$R_{\mathcal{D}}(Q) \leq \hat{R}_{\mathcal{S}_m}(Q) + \frac{G}{2} W_2^2(Q, P) + \sqrt{\frac{2\sigma^2 \log\left(\frac{1}{\delta}\right)}{m}} + D \underset{h \sim Q}{\mathbb{E}} \left( \left\| \nabla_h R_{\mathcal{D}}(h) - \nabla_h \hat{R}_{\mathcal{S}_m}(h) \right\| \right).$$

The proof is deferred to Appendix C.7. Theorem 14 is not the first generalisation bound to involve a 2-Wasserstein distance (Lugosi and Neu, 2022, 2023). However, these results require infinitely smooth loss functions. Additionally, the results from Amit et al. (2022); Haddouche and Guedj (2023b); Viallard et al. (2023b), which use the 1-Wasserstein, can be directly relaxed on bounds involving the 2-Wasserstein, while still requiring a Lipschitz loss. In contrast, our result holds for any nonnegative loss whom the generalisation gap $h \to R_{\mathcal{D}}(h) - \hat{R}_{\mathcal{S}_m}(h)$, satisfies $(\star)$, being a relaxation of gradient-Lipschitz assumption. Theorem 14 involves a rate of $1/\sqrt{m}$, as we have to control the generalisation gap over P. Another restriction of our result, compared to previous ones, is that it holds for $\mathcal{H}$ with a finite diameter.

**Can Theorem 14 go to zero with large $m$?** In its current form, it is unclear whether Theorem 14 goes to zero with large $m$, as the Wasserstein distance and the gradient term, have no explicit convergence rate in $m$. Concerning the gradient term, it has been shown that, if we consider Q to be a Dirac in the output of SGD after $T$ iterations, then according to Li and Liu (2022, Theorem 3), when the intrinsic noise of SGD is subgaussian, the output $\mathbf{w}_T \in \mathbb{R}^d$ of SGD satisfies with high probability: $\|\nabla_h R_{\mathcal{S}_m}(\mathbf{w}_T) - \nabla_h R_{\mathcal{D}}(\mathbf{w}_T)\|^2 \leq \mathcal{O}\left( \frac{d\sqrt{T} \log(T)}{m} \right)$. Concerning the Wasserstein term, if we assume directly that $\ell$ satisfies $(\star)$ with constant $G$, then using a technique inspired from Amit et al. (2022) yields with probability $1 - \delta$ that the generalisation gap satisfies $(\star)$ with constant $G' = \mathcal{O}\left( G\sqrt{\log(|\mathcal{H}|/\delta)/m} \right)$ when $\mathcal{H}$ is finite. We prove this in Appendix A.5.

## 6. An empirical study of Assumption 5 for neural networks[2]

In this section, we empirically verify whether the QSB assumption holds for neural nets. This allows us to assess whether Theorem 6 helps in understanding the generalisation ability of neural nets.

**Experimental protocol.** We consider classification tasks on two datasets: MNIST (LeCun, 1998) and FashionMNIST (Xiao et al., 2017). We have kept the original training set $\mathcal{S}_m$ and the original test set denoted by $\mathcal{T}_n$ (of size $n$). We consider the convolutional neural network of Springenberg et al. (2015) adapted for MNIST and FashionMNIST. The model is composed of 4 layers containing 10 channels with a $5 \times 5$-kernel; we set the stride and the padding to 1, except for the second layer, where it is fixed to 2. Each of these (convolutional) layers is followed by a Leaky ReLU activation function. Moreover, an average pooling with a $8 \times 8$-kernel is performed before the Softmax activation function. To initialise the weights of the network, we use Glorot and Bengio (2010) uniform initialiser, while the biases are initialised in $[-1/\sqrt{250}, +1/\sqrt{250}]$ uniformly (except the first layer, the interval is $[-1/5, +1/5]$). Hence, in this case, $\mathcal{H}$ is the set of neural networks with a fixed architecture, and parametrised with a vector $\mathbf{w}$. The posterior distribution Q is a Gaussian measure $\mathcal{N}(\mathbf{w}, \sigma^2 \mathrm{Id})$ centred on the parameters $\mathbf{w}$ associated with the model; $\sigma$ is set to $10^{-4}$. Note that this distribution

---

2. The source code is available at this link.

respects the $\text{Poinc}(c_P)$ assumption; see Section 3.1. We train the neural network with the (vanilla) stochastic gradient descent algorithm, where the batch size is equal to 512, and the learning rate is fixed to $10^{-2}$. We train for at least $10^4$ gradient steps and finish the current epoch when this number of iterations is reached. Our loss $\ell$ is the bounded cross-entropy loss of Dziugaite and Roy (2017, Section D).

In Figure 1, we report the evolution of three quantities: *(i)* the estimated value of $C$, *(ii)* the test risk $\hat{R}_{\mathcal{T}_n}(Q)$ and *(iii)* the test risk with the 01-loss. More precisely, the risks and $C$ are estimated by sampling 10 hypotheses from $Q$ and by computing the values on a mini-batch of $\mathcal{T}_n$ (with 512 examples) at each iteration. Then, Figure 1 represents averaged values on 5 runs, each point of the curve representing the average on 100 iterations of the training process (for $10^4$ iterations, we only plot $10^2$ averaged points for clarity).

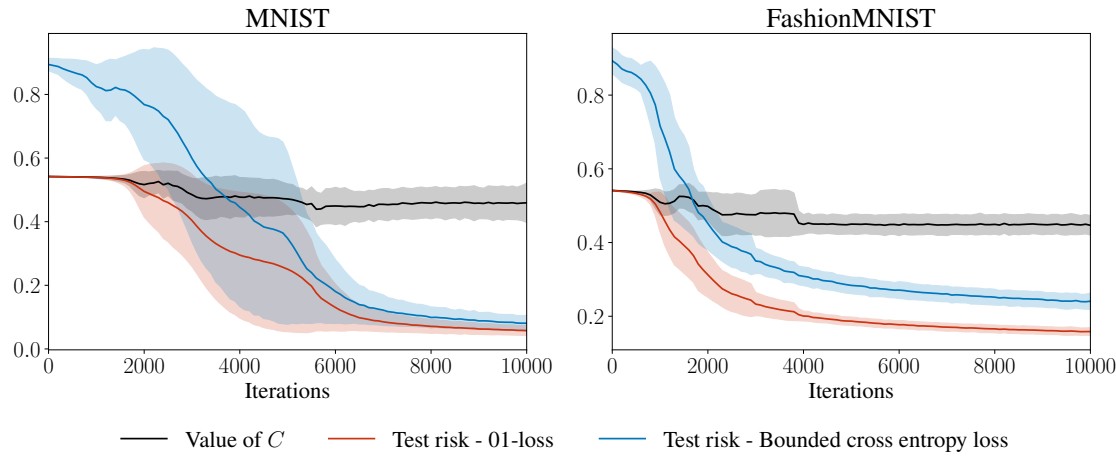

Figure 1: Evolution of the test risks (with the 01-loss and the bounded cross-entropy loss) and the value of $C$ during the training phase.

**Empirical findings.** Figure 1 illustrates that, when neural networks are involved for two classification tasks, $Q$ evolves during the optimisation process while maintaining the QSB property with constant $C < 1$. For both MNIST and FashionMNIST, the constant $C$ decreases from approximately 0.55 to 0.45. We deduce that having a data-free $P$ (0 iteration) being QSB with $C < 1$ suggests that the architecture of our neural network also has an influence on the QSB assumption. As specified in Section 3, having $C < 1$ attenuates the impact of the KL term, thus $P$. This is desirable as it allows the optimiser to deeply explore the predictor space when $P$ yields poor performances. We also note that the generalisation ability of $Q$ on the training loss nearly matches the performance on the 0-1 loss for MNIST but is deteriorated for FashionMNIST, this invites to study more deeply the design of such surrogates in future work.

Finally, the take-home message of this study is that the QSB assumption is verified for small neural networks on MNIST. Such an empirical confirmation is crucial as it is required for our main

result (Theorem 6) and thus confirms that, for neural networks, reaching flat minima during the optimisation phase translates in increased generalisation ability.

## 7. Conclusion

We provide novel time-uniform PAC-Bayes bounds, that can be interpreted as a transitory convergence rate of $1/m$ when a low empirical error is reached and expected gradients vanish. Doing so, we draw sound theoretical links highlighting the impact of flat minima in generalisation. However, a crucial open question remains: how do optimisation algorithms successfully attain flat minima in the overparameterised setting? We leave this important question for future work.

## Acknowledgments

M. Haddouche and U. Şimşekli are supported by the European Union (ERC grant DYNASTY 101039676). The French government partly funded this work under the management of Agence Nationale de la Recherche as part of the "France 2030" program, reference ANR-23-IACL-0008 (PR[AI]RIE-PSAI). B. Guedj acknowledges partial support from the French National Agency for Research, through the programme "France 2030" and PEPR IA on grant SHARP ANR-23-PEIA-0008. We would like to thank all the reviewers for the fruitful discussion phase, which greatly enhanced this work.

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

## Appendix A. Supplementary details

### A.1. Additional details on Poincaré and Log-Sobolev inequalities

We first provide a proof of Proposition 4.

**Proposition 4** *Assume that* $P$ *is a probability measure on* $\mathcal{H}$ *such that* $dP(h) \propto \exp(-V(h))$ *with* $V : \mathcal{H} \to \mathbb{R}$ *a smooth function such that* $\text{Hess}(V) \succeq \frac{2}{c_{LS}(P)}\text{Id}$.[3] *Assume that* $\ell(h, \mathbf{z}) = \ell_1(h, \mathbf{z}) + \ell_2(h, \mathbf{z})$ *with* $\ell_1$ *convex, twice differentiable and* $\ell_2$ *bounded. Then for any* $\gamma > 0$*, the Gibbs posterior* $Q_{\mathcal{S}_m}^{\gamma}$ *is* L-Sob$(c_{LS})$ *with constant* $c_{LS}(Q_{\mathcal{S}_m}^{\gamma}) = c_{LS}(P)\exp(4\|\ell_2\|_\infty)$.

**Proof** We define $P_1$ such that $dP_1(h) \propto \exp\left(-V(h) - \frac{\gamma}{m}\sum_{i=1}^{m}\ell_1(h, \mathbf{z}_i)\right)dh$. Then, by the convexity assumption over the loss $\ell_1$, we have $\text{Hess}(V + \frac{\gamma}{m}\sum_{i=1}^{m}\ell_1(h, \mathbf{z}_i)) \succeq \frac{1}{c_{LS}}\text{Id}$. Then, applying Chafaï (2004, Corollary 2.1), we know that $P_1$ satisfies a Poincaré inequality with constant $c_{LS}(P)$. Finally, defining $P_2$ such that $dP_2(h) \propto \exp\left(-\frac{\gamma}{m}\sum_{i=1}^{m}\ell_2(h, \mathbf{z}_i)\right)dh$, thanks to the boundedness of the loss $\ell_2$, we use Guionnet and Zegarlinksi (2003, Property 4.6), which ensures that $dP_2(h) = Q_{\mathcal{S}_m}^{\gamma}dP_1(h)$ satisfies a Log-Sobolev inequality with constant $c_{LS}(P)\exp(4\|\ell_2\|_\infty)$. Observing that $P_2 = Q_{\mathcal{S}_m}^{\gamma}$ completes the proof. ∎

For the sake of completeness, we prove Proposition 15, which is Proposition 2.1 of Ledoux (2006), showing that a Log-Sobolev inequality implies a Poincaré inequality.

**Proposition 15 (Proposition 2.1 of Ledoux (2006))** *If the distribution* $Q$ *is* L-Sob$(c_{LS})$*, then it is also* Poinc$(c_P)$*, with* $c_P(Q) = \frac{c_{LS}(Q)}{2}$.

**Proof** Let $f \in H^1(Q)$ such that $\mathbb{E}_{h \sim Q}[f(h)] = 0$ and $\mathbb{E}_{h \sim Q}[f^2(h)] = 1$. For any $\varepsilon > 0$, we have $1 + \varepsilon f \in H^1(Q)$. We then apply the Log-Sobolev inequality on $1 + \varepsilon f$ to obtain

$$\mathop{\mathbb{E}}_{h \sim Q}\left[(1 + \varepsilon f(h))^2\left(2\log(1 + \varepsilon f(h)) - \log(1 + \varepsilon^2)\right)\right] \leq c_{LS}(Q)\varepsilon^2 \mathop{\mathbb{E}}_{h \sim Q}\left[\|\nabla_h f(h)\|^2\right].$$

Note that, by a Taylor expansion, we have $\log(1 + \varepsilon f(h)) = \varepsilon f(h) - \frac{(\varepsilon f(h))^2}{2} + o(\varepsilon^2)$ and we have also $\log(1 + \varepsilon^2) = \varepsilon^2 + o(\varepsilon^2)$. Then, plugging this into the previous equation gives

$$\mathop{\mathbb{E}}_{h \sim Q}\left[2\varepsilon f(h) + 3(\varepsilon f(h))^2 - \varepsilon^2 + o(\varepsilon^2)\right] \leq c_{LS}(Q)\varepsilon^2 \mathop{\mathbb{E}}_{h \sim Q}\left[\|\nabla_h f(h)\|^2\right].$$

We use that $\mathbb{E}_{h \sim Q}[f(h)] = 0$ and we then divide by $\varepsilon^2$. Taking the limit $\varepsilon \to 0$ gives:

$$\mathop{\mathbb{E}}_{h \sim Q}\left[3f(h)^2 - 1\right] \leq c_{LS}(Q) \mathop{\mathbb{E}}_{h \sim Q}\left[\|\nabla_h f(h)\|^2\right].$$

Using that $\mathbb{E}_{h \sim Q}[f^2(h)] = 1$, we obtain

$$1 \leq \frac{c_{LS}(Q)}{2} \mathop{\mathbb{E}}_{h \sim Q}\left[\|\nabla_h f(h)\|^2\right].$$

Then, for any $g \in H^1(Q)$ applying this proof on $f : h \mapsto \frac{g(h) - \mathbb{E}_{h \sim Q}[g(h)]}{\sqrt{\text{Var}_{h \sim Q}(g(h))}}$ concludes the proof. ∎

---

3. The notation $A \succeq B$ means that $A - B$ is a semi-definite positive matrix.

## A.2. Wasserstein distances

We recall the definitions of the 1-Wasserstein and the 2-Wasserstein distances, which are valid for any predictor space $\mathcal{H} \subseteq \mathbb{R}^d$ equipped with the Euclidean distance.

**Definition 16** *The* 1*-Wasserstein distance between* $Q \in \mathcal{M}(\mathcal{H})$ *and* $P \in \mathcal{M}(\mathcal{H})$ *is defined as*

$$W_1(Q, P) = \inf_{\pi \in \Pi(Q, P)} \int_{\mathcal{H}^2} \|x - y\| d\pi(x, y).$$

*where* $\Pi(Q, P)$ *denotes the set of probability measures on* $\mathcal{H}^2$ *whose marginals are* $Q$ *and* $P$. *Similarly, the* 2*-Wasserstein distance between* $Q \in \mathcal{M}(\mathcal{H})$ *and* $P \in \mathcal{M}(\mathcal{H})$ *is defined as*

$$W_2(Q, P) = \sqrt{\inf_{\pi \in \Pi(Q, P)} \int_{\mathcal{H}^2} \|x - y\|^2 d\pi(x, y)}.$$

## A.3. Fundamental difference between time-uniform PAC-Bayes bounds and classical ones

In this work, we establish *time-uniform estimation* PAC-Bayes bounds. Specifically, we focus on the bounds such that there exist $\mathcal{C} \subseteq \mathcal{M}(\mathcal{H})$, a threshold $\varepsilon > 0$, and $\alpha > 0$ such that, with probability at least $1 - \delta$, for all $Q \in \mathcal{C}$ and $m \in \mathbb{N}^*$, we have

$$R_{\mathcal{D}}(Q) \leq \frac{\alpha}{m} \left[ KL(Q, P) + \log(1/\delta) \right] + \varepsilon_m.$$

We show that, in a favourable setting, time-uniform estimation PAC-Bayes bounds are sufficient to ensure the almost sure convergence of $(R_{\mathcal{D}}(Q_m))_{m \in \mathbb{N}^*}$ for a posterior sequence $(Q_m)_{m \in \mathbb{N}^*}$, whereas classical PAC-Bayes bounds provide only convergence in probability.

Let us first consider a nonnegative loss $\ell$, a prior $P$, a countable dataset $\mathcal{S}$, and assume there exists a sequence $(Q_m)_{m \in \mathbb{N}^*}$ which is such that $KL(Q_m, P) \leq D$ for all $m \in \mathbb{N}^*$ and both $\hat{R}_{\mathcal{S}_m}(Q_m)$ and $\mathbb{E}_{\mathbf{z} \sim \mathcal{D}, h \sim Q_m} \left[ \|\nabla_h \ell(h, \mathbf{z})\|^2 \right]$ go to zero as $m$ goes to infinity almost surely. Then, the classical McAllester's bound states that for any $m \in \mathbb{N}^*$, and $\delta_m \in (0, 1]$, with probability at least $1 - \delta_m$ over $\mathcal{S}_m$, we have

$$\mathbb{P}_{\mathcal{S}_m} \left( R_{\mathcal{D}}(Q_m) \leq \hat{R}_{\mathcal{S}_m}(Q_m) + \sqrt{\frac{KL(Q_m, P) + \log(2\sqrt{m}/\delta_m)}{2m}} \right) \geq 1 - \delta_m.$$

Let $\alpha > 0$ and take for all $m \in \mathbb{N}^*$, the confidence parameter $\delta_m = \delta/m$. Since we assume that $\hat{R}_{\mathcal{S}_m}(Q_m) \to 0$, the square root in McAllester's bound goes to zero. Thus, there exists $m_0 \in \mathbb{N}^*$ such that for all $m \geq m_0$, we have

$$R_{\mathcal{D}}(Q_m) \leq \hat{R}_{\mathcal{S}_m}(Q_m) + \sqrt{\frac{KL(Q_m, P) + \log(2m\sqrt{m}/\delta)}{2m}} \leq \alpha.$$

Thus, for all $m \geq m_0$, we know that $\mathbb{P}_{\mathcal{S}_m} (R_{\mathcal{D}}(Q_m) < \alpha) \geq 1 - \delta/m$. Taking the limit as $m$ goes to infinity ensures that for any $\alpha > 0$, we have

$$\lim_{m \to \infty} \mathbb{P}_{\mathcal{S}_m} (R_{\mathcal{D}}(Q_m) < \alpha) = 1.$$

Thus, the classical McAllester's bound allows the sequence $R_{\mathcal{D}}(Q_m)$ to converge in probability to zero. In contrast, Theorem 6 with $\lambda = 1/C$ is a time-uniform estimation bound with $\alpha = 2C$ and $\varepsilon_m = 2\left(\hat{R}_{\mathcal{S}_m}(Q) + \frac{1}{2C}\mathbb{E}_{\mathbf{z}\sim\mathcal{D}, h\sim Q}\left[\|\nabla_h\ell(h,\mathbf{z})\|^2\right]\right)$. Under our assumptions, we know that both $\frac{\alpha}{m}\left[\mathrm{KL}(Q,P) + \log(1/\delta)\right]$ and $\varepsilon_m$ go to zero as $m$ goes to infinity. Then, taking the limit, we have for all $\delta > 0$

$$\mathbb{P}_{\mathcal{S}}\left(\lim_{m\to\infty} R_{\mathcal{D}}(Q_m) = 0\right) \geq 1 - \delta.$$

As $\lim_{m\to\infty} R_{\mathcal{D}}(Q_m)$ does not depend on $\delta$, we can then make $\delta$ go to zero to obtain

$$\mathbb{P}_{\mathcal{S}}\left(\lim_{m\to\infty} R_{\mathcal{D}}(Q_m) = 0\right) = 1.$$

Thus, $R_{\mathcal{D}}(Q_m)$ converges almost surely to zero. Such a conclusion cannot be achieved with classical PAC-Bayes bounds; our bound is stronger as it allows an almost sure convergence, which cannot be obtained by simple manipulations of classical bounds.

### A.4. A deeper look at the nature of our gradient terms and their originality in PAC-Bayes

Here, we highlight the novelty of incorporating $\nabla_h\ell(h,\mathbf{z})$ in PAC-Bayes bounds and argue that this is not a trivial extension of the PAC-Bayes Bernstein bound. Given a bounded loss $\ell : \mathcal{H} \times \mathcal{Z} \to [0,1]$, and starting from the Bernstein bound of Tolstikhin and Seldin (2013), it is possible to derive a bound involving gradient terms with an additional cost when the posterior satisfies a Poincaré inequality. We first restate the PAC-Bayes Bernstein bound of Tolstikhin and Seldin (2013).

**Theorem 17** *For any $c_1 > 1$, for any data-free prior $P \in \mathcal{M}(\mathcal{H})$, for any loss function $\ell : \mathcal{H} \times \mathcal{Z} \to [0,1]$, and for any $\delta \in (0,1]$, with probability at least $1 - \delta$ over the sample $\mathcal{S}$, we have for all distributions $Q \in \mathcal{M}(\mathcal{H})$ that satisfy*

$$\sqrt{\frac{\mathrm{KL}(Q,P) + \ln\frac{\nu_1}{\delta_1}}{(e-2)\,\mathbb{E}_{h\sim Q}[\mathrm{Var}_{\mathbf{z}\sim\mathcal{D}}(\ell(h,\mathbf{z}))]}} \leq \sqrt{m}, \tag{3}$$

*we have*

$$R_{\mathcal{D}}(Q) \leq \hat{R}_{\mathcal{S}_m}(Q) + (1+c_1)\sqrt{\frac{(e-2)\,\mathbb{E}_{h\sim Q}[\mathrm{Var}_{\mathbf{z}\sim\mathcal{D}}(\ell(h,\mathbf{z}))]\left(\mathrm{KL}(Q,P) + \ln\frac{\nu_1}{\delta}\right)}{m}},$$

*where*

$$\nu_1 = \left\lceil \frac{1}{\ln c_1}\ln\left(\sqrt{\frac{(e-2)m}{4\ln(1/\delta)}}\right)\right\rceil + 1.$$

Assuming the technical conditions of Theorem 17, that the distribution $\mathcal{D}$ satisfies a Poincaré inequality, and that the loss $\ell(h,\cdot) \in \mathrm{H}^1(\mathcal{D})$ for all $h \in \mathcal{H}$, we obtain the following corollary.

**Corollary 18** *Under the same conditions of Theorem 17, for any distribution $\mathcal{D}$ being $\mathrm{Poinc}(c_{\mathcal{D}})$, and for any loss $\ell(h,\cdot) \in \mathrm{H}^1(\mathcal{D})$, with probability at least $1 - \delta$ over the sample $\mathcal{S}$, we have for all distributions $Q \in \mathcal{M}(\mathcal{H})$ satisfying Equation (3)*

$$R_{\mathcal{D}}(Q) \leq \hat{R}_{\mathcal{S}_m}(Q) + (1+c_1)\sqrt{\frac{(e-2)\,\mathbb{E}_{h\sim Q}\,\mathbb{E}_{\mathbf{z}\sim\mathcal{D}}\left[\|\nabla_{\mathbf{z}}\ell(h,\mathbf{z})\|^2\right]\left(\mathrm{KL}(Q,P) + \ln\frac{\nu_1}{\delta}\right)}{m}}.$$

**Proof** From Poincaré inequality (Definition 1), we have

$$\mathop{\mathbb{E}}_{h \sim Q} \left[ \mathop{\mathrm{Var}}_{\mathbf{z} \sim \mathcal{D}} (\ell(h, \mathbf{z})) \right] \leq \mathop{\mathbb{E}}_{h \sim Q} \mathop{\mathbb{E}}_{\mathbf{z} \sim \mathcal{D}} \left[ \|\nabla_{\mathbf{z}} \ell(h, \mathbf{z})\|^2 \right],$$

which is then substituted into the bound of Theorem 17. ∎

By using the additional Poincaré assumption on the data distribution $\mathcal{D}$, we can obtain a bound with gradient term $\mathbb{E}_{h \sim Q} \mathbb{E}_{\mathbf{z} \sim \mathcal{D}}[\|\nabla_{\mathbf{z}} \ell(h, \mathbf{z})\|^2]$. However, it remains unclear how $\nabla_{\mathbf{z}} \ell(h, \mathbf{z})$ behaves, as we do not optimise with respect to $\mathbf{z}$. As a result, these gradients may remain large even after a successful learning phase. In contrast, Theorem 6 involves the term $\mathbb{E}_{\mathbf{z} \sim \mathcal{D}} \mathbb{E}_{h \sim Q}[\|\nabla_h \ell(h, \mathbf{z})\|^2]$, which is minimised during a successful optimisation process. Moreover, the Poincaré assumption on the data distribution $\mathcal{D}$ is difficult to verify, as we do not know $\mathcal{D}$. Our Poincaré assumption on the posterior Q is much easier to verify, as we can choose Q in practice. A similar discussion then applies for the empirical Bernstein bound (Tolstikhin and Seldin, 2013, Theorem 4) as the empirical variance is with respect to the dataset $\mathcal{S}$.

### A.5. How to make the Wasserstein term go to zero in Theorem 14?

In this section, we prove the following lemma.

**Lemma 19** *Assume that the hypothesis set $\mathcal{H}$ is finite and that the loss $\ell : \mathcal{H} \times \mathcal{Z} \to \mathbb{R}$ satisfies $(\star)$ with constant $G$. Then, with probability at least $1 - \delta$, the generalisation gap $\mathrm{R}_{\mathcal{D}}(h) - \hat{\mathrm{R}}_{\mathcal{S}_m}(h)$ satisfies $(\star)$ with the constant $G' = \mathcal{O}\left( G\sqrt{\log(|\mathcal{H}|/\delta)/m} \right)$.*

**Proof** For the sake of simplicity, let $f : h \mapsto \mathrm{R}_{\mathcal{D}}(h) - \hat{\mathrm{R}}_{\mathcal{S}_m}(h)$ be the generalisation gap. Then, fix $(h_1, h_2, a, a') \in \mathcal{H}^4$ and our goal is to prove that $f$ satisfies $(\star)$ with another constant $G'$. First of all, notice that

$$\langle \nabla_h f(h_1) - \nabla_h f(h_2), a - a' \rangle$$
$$= \frac{1}{m} \sum_{i=1}^{m} \langle \nabla_h \ell(h_2, \mathbf{z}_i) - \nabla_h \ell(h_1, \mathbf{z}_i) - (\nabla_h \mathrm{R}_{\mathcal{D}}(h) - \nabla_h \mathrm{R}_{\mathcal{D}}(h)), a - a' \rangle.$$

Moreover, by the condition $(\star)$ for all $\mathbf{z} \in \mathcal{Z}$, we know that

$$|\langle \nabla_h \ell(h_2, \mathbf{z}) - \nabla_h \ell(h_1, \mathbf{z}), a - a' \rangle| \leq G \cdot \|a - a'\| \cdot \|h_1 - h_2\|.$$

Then, by Hoeffding's inequality, applied on the centered random variable $\nabla_h \ell(h_2, \mathbf{z}_i) - \nabla_h \ell(h_1, \mathbf{z}_i) - (\nabla_h \mathrm{R}_{\mathcal{D}}(h_2) - \nabla_h \mathrm{R}_{\mathcal{D}}(h_1))$ bounded by $G\|a - a'\|\|h_1 - h_2\|$, with probability at least $1 - \delta$, we have

$$\left| \langle \nabla_h f(h_1) - \nabla_h f(h_2), a - a' \rangle \right| \leq G \cdot \|a - a'\| \cdot \|h_1 - h_2\| \cdot \sqrt{\frac{2 \log(2/\delta)}{m}}.$$

Taking a union bound on all possible values of $(h_1, h_2, a, a') \in \mathcal{H}^4$ with $\delta' = \delta/|\mathcal{H}|^4$ and a union bound on all tuples yields that, with probability at least $1 - \delta$, for all $(a, a') \in \mathcal{H}^2$,

the function $h \mapsto \langle \nabla_h f(h), a - a' \rangle$ is $G\sqrt{\dfrac{2 \log \left( \frac{2|\mathcal{H}|^4}{\delta} \right)}{m}} \|a - a'\|$-Lipschitz,

meaning the condition $(\star)$ is verified with constant $G' = G\sqrt{\frac{2\log\left(\frac{2|\mathcal{H}|^4}{\delta}\right)}{m}}$ for the gap. ∎

## Appendix B. PAC-Bayes bounds for Lipschitz losses through log-Sobolev inequalities

**Extending Catoni's bound to Lipschitz losses.** A well-known relaxation of Catoni (2007, Theorem 1.2.6) (see also Alquier et al., 2016, Theorem 4.1) holding for subgaussian losses has been widely used in practice as a tractable PAC-Bayesian algorithm exhibiting a linear dependency on the KL divergence. Below, we exploit a consequence of the Herbst argument, as stated, for example, in Ledoux (2006, Section 2.3), which asserts that an $L$-Lipschitz function of a random variable following a distribution $\mathcal{D}$ being $\mathtt{L-Sob}(c_{LS})$ is $L\sqrt{c_{LS}(\mathcal{D})}$-subgaussian. This leads to the following corollary.

**Corollary 20** *For any $\lambda > 0$, for any data-free prior $\mathrm{P} \in \mathcal{M}(\mathcal{H})$, for any $L$-Lipschitz loss $\ell$ : $\mathcal{H} \times \mathcal{Z} \to \mathbb{R}$ for any $h \in \mathcal{H}$, for any data distribution $\mathcal{D}$ being $\mathtt{L-Sob}(c_{LS})$, with probability at least $1 - \delta$ over $\mathcal{S}$, for any $\mathrm{Q} \in \mathcal{M}(\mathcal{H})$, we have*

$$\mathrm{R}_{\mathcal{D}}(\mathrm{Q}) \leq \hat{\mathrm{R}}_{\mathcal{S}_m}(\mathrm{Q}) + \frac{\mathrm{KL}(\mathrm{Q},\mathrm{P}) + \log(1/\delta)}{\lambda} + \frac{\lambda^2 L^2 c_{LS}(\mathcal{D})}{2m}.$$

**Proof** First, we take $f(h, \mathcal{S}_m) := \lambda(\mathrm{R}_{\mathcal{D}}(h) - \hat{\mathrm{R}}_{\mathcal{S}_m}(h))$ and we use the change of measure inequality (Csiszár, 1975; Donsker and Varadhan, 1976) to state that, for all $\mathrm{Q} \in \mathcal{M}(\mathcal{H})$, we have

$$\mathop{\mathbb{E}}_{h\sim\mathrm{Q}}[f(h, \mathcal{S}_m)] \leq \mathrm{KL}(\mathrm{Q},\mathrm{P}) + \log\left(\mathop{\mathbb{E}}_{h\sim\mathrm{P}}\left[\exp\left(f(h, \mathcal{S}_m)\right)\right]\right).$$

Moreover, Markov's inequality alongside Fubini's theorem gives, with probability at least $1 - \delta$ over the sample $\mathcal{S}$,

$$\mathop{\mathbb{E}}_{h\sim\mathrm{Q}}[f(h, \mathcal{S}_m)] \leq \mathrm{KL}(\mathrm{Q},\mathrm{P}) + \log(1/\delta) + \log\left(\mathop{\mathbb{E}}_{h\sim\mathrm{P}}\mathop{\mathbb{E}}_{\mathcal{S}_m}\left[\exp\left(f(h, \mathcal{S}_m)\right)\right]\right).$$

Now, since the loss $\ell$ is $L$-Lipschitz on $\mathbf{z} \in \mathcal{Z}$ for all $h \in \mathcal{H}$, we show below that the function $f$ is $\frac{\lambda L}{\sqrt{m}}$-Lipschitz on the variable $\mathcal{S}_m$ for each $h \in \mathcal{H}$. Indeed, as the loss is $L$-Lipschitz w.r.t. $\mathbf{z} \in \mathcal{Z}$, for any dataset $\mathcal{S}_m = (\mathbf{z}_1, \ldots, \mathbf{z}_m)$, any $\mathcal{S}'_m = (\mathbf{z}'_1, \cdots, \mathbf{z}'_m)$, and any $h \in \mathcal{H}$, we have

$$\|f(h, \mathcal{S}_m) - f(h, \mathcal{S}'_m)\| \leq \frac{\lambda L}{m}\sum_{i=1}^{m}\|\mathbf{z}_i - \mathbf{z}'_i\| = \frac{\lambda L}{m}\sum_{i=1}^{m}\sqrt{\|\mathbf{z}_i - \mathbf{z}'_i\|^2}.$$

Then, by the concavity of the square root, we have

$$\|f(h, \mathcal{S}_m) - f(h, \mathcal{S}'_m)\| \leq \lambda L\sqrt{\frac{1}{m}\sum_{i=1}^{m}\|\mathbf{z}_i - \mathbf{z}'_i\|^2} = \lambda L\frac{\|\mathcal{S}_m - \mathcal{S}'_m\|}{\sqrt{m}}.$$

The underlying norm $\|\mathcal{S}'_m\|$ is the one derived from the scalar product $\langle\mathcal{S}_m, \mathcal{S}'_m\rangle = \sum_{i=1}^{m}\langle\mathbf{z}_i, \mathbf{z}'_i\rangle$. As the distribution $\mathcal{D}$ is $\mathtt{L-Sob}(c_{LS})$, we can deduce that $\mathcal{D}^{\otimes m}$ is also $\mathtt{L-Sob}(c_{LS})$ with the same

constant (Ané et al., 2000, Corollary 3.2.3). Then, using the Herbst argument similarly to Ledoux (2006, Section 2.3), we conclude that for all $h \in \mathcal{H}$, the function $f(h, \cdot)$ is $L\lambda\sqrt{\frac{c_{LS}(\mathcal{D})}{m}}$-subgaussian. Thus, we have

$$\log\left(\mathop{\mathbb{E}}_{h \sim \mathrm{P}} \mathop{\mathbb{E}}_{\mathcal{S}_m} \left[\exp\left(f(h, \mathcal{S}_m)\right)\right]\right) \leq \frac{\lambda^2 L^2 c_{LS}(\mathcal{D})}{2m},$$

which concludes the proof. ∎

**Disintegrated PAC-Bayes bounds.** Numerical estimation of PAC-Bayes bounds is usually challenging as it often involves Monte-Carlo approximations of the expectation over the posterior Q. A recent line of work (Rivasplata et al., 2020; Haddouche and Guedj, 2022; Viallard et al., 2023a, 2024a) studies *disintegrated PAC-Bayes bounds*, *i.e.* bounds that hold with high probability on both the dataset $\mathcal{S}$ and a single predictor $h$ drawn from the posterior Q. These bounds may be relevant for practitioners when sampling is easy, as in the case of Gaussian distributions, since they require little computational time. However, a drawback of these bounds is that they do not allow the KL divergence to be used as a complexity measure. Instead, either the disintegrated KL (Rivasplata et al., 2020) or the Rényi divergence (Viallard et al., 2023a) is considered, which can be seen as a relaxation of the KL divergence one. By leveraging the subgaussianity behaviour of Lipschitz losses, it is possible to derive PAC-Bayesian disintegrated bounds, as long as the posterior distribution satisfies a log-Sobolev inequality with a sharp constant (which can be achieved, for instance, for Gaussian distributions with a small operator norm). The new disintegrated bound is introduced in the following lemma.

**Lemma 21** *For any $L$-Lipschitz loss $\ell : \mathcal{H} \times \mathcal{Z} \to \mathbb{R}$, for any distribution Q being $L-Sob(c_{LS})$ with $c_{LS}(\mathrm{Q}) \leq 1/m$ (and that can depend on the dataset $\mathcal{S}_m$), with probability $1 - \delta$ over the draw of $h \sim \mathrm{Q}$ and $\mathcal{S}_m$, we have*

$$\mathrm{R}_{\mathcal{D}}(h) - \hat{\mathrm{R}}_{\mathcal{S}_m}(h) \leq \mathrm{R}_{\mathcal{D}}(\mathrm{Q}) - \hat{\mathrm{R}}_{\mathcal{S}_m}(\mathrm{Q}) + \sqrt{\frac{L^2 \log(1/\delta)}{2m}}.$$

**Proof** We simply remark, by the same argument as in the proof of Corollary 20 that the gap $h \mapsto \mathrm{R}_{\mathcal{D}}(h) - \mathrm{R}_{\mathcal{S}_m}(h)$ is $L$-Lipschitz for any $\mathcal{S}_m$ thus the gap is $L\sqrt{c_{LS}(\mathrm{Q})}$-subgaussian. Then we use

$$\mathrm{R}_{\mathcal{D}}(h) - \hat{\mathrm{R}}_{\mathcal{S}_m}(h) = \log\left(\exp\left(\mathrm{R}_{\mathcal{D}}(h) - \hat{\mathrm{R}}_{\mathcal{S}_m}(h)\right)\right).$$

We then apply Markov inequality and exploit the subgaussiannity of the gap alongside $c_{LS}(\mathrm{Q}) \leq 1/m$ to conclude the proof. ∎

This lemma states that, as long as we assume our loss to be Lipschitz with respect to $h$, it is possible to easily derive disintegrated PAC-Bayesian bounds. Additionally, Lemma 21 can be easily completed by Corollary 20, which introduces a KL divergence as a complexity term. Note also that, since the loss is Lipschitz, it is also possible to incorporate the 1-Wasserstein distance through the bounds of Haddouche and Guedj (2023b); Viallard et al. (2023b, 2024b). Therefore, having a Log-Sobolev assumption with a sharp constant on the posterior distribution is enough to provide disintegrated PAC-Bayesian bounds involving the KL divergence or the Wasserstein distance, rather than the Rényi divergence or the disintegrated KL divergence.

## Appendix C. Proofs

### C.1. Proof of Corollary 7

The goal of this section is to prove Corollary 7, which is restated for ease of readability.

**Corollary 7** *For any $C>0$, for any $\lambda$ such that $\frac{2}{C}>\lambda>0$, for any data-free prior $\mathrm{P} \in \mathcal{M}(\mathcal{H})$, for any loss function $\ell : \mathcal{H} \times \mathcal{Z} \to \mathbb{R}_+$ such that $\ell(h,\cdot)$ is $\mathcal{C}^1$ almost everywhere on $\mathcal{Z}$, any $\delta \in (0,1]$, if the data distribution $\mathcal{D}$ is $\texttt{Poinc}(c_{\mathrm{P}})$, with probability at least $1 - \delta$ over the sample $\mathcal{S}$, for any $m \in \mathbb{N}^*$, any posterior $\mathrm{Q}$ being $\texttt{Poinc}(c_{\mathrm{P}})$ with $\mathrm{R}_\mathcal{D}(\mathrm{Q}) \leq C$ and such that for any $\mathbf{z} \in \mathcal{Z}$, $\ell(\cdot, \mathbf{z}) \in \mathrm{H}^1(\mathrm{Q})$:*

$$
\mathrm{R}_\mathcal{D}(\mathrm{Q}) \leq \frac{1}{1 - \frac{\lambda C}{2}} \left( \hat{\mathrm{R}}_{\mathcal{S}_m}(\mathrm{Q}) + \frac{\mathrm{KL}(\mathrm{Q},\mathrm{P}) + \log(1/\delta)}{\lambda m} \right)
$$
$$
+ \frac{\lambda}{2 - \lambda C} \left( c_P(\mathrm{Q}) \underset{\mathbf{z}\sim\mathcal{D}}{\mathbb{E}} \left[ \underset{h\sim\mathrm{Q}}{\mathbb{E}} \left( \|\nabla_h \ell(h,\mathbf{z})\|^2 \right) \right] + c_P(\mathcal{D}) \underset{\mathbf{z}\sim\mathcal{D}}{\mathbb{E}} \left( \left\| \underset{h\sim\mathrm{Q}}{\mathbb{E}}[\nabla_{\mathbf{z}} \ell(h,\mathbf{z})] \right\|^2 \right) \right).
$$

To begin this proof, we first state an important intermediate theorem, which holds without any assumptions on the data distribution.

**Theorem 22** *For any $C > 0$, for any $\lambda$ such that $\frac{2}{C}>\lambda>0$, for any data-free prior $\mathrm{P} \in \mathcal{M}(\mathcal{H})$, for any loss function $\ell : \mathcal{H} \times \mathcal{Z} \to \mathbb{R}_+$, and for any $\delta \in (0,1]$, with probability at least $1 - \delta$ over the sample $\mathcal{S}$, for all $m \in \mathbb{N}^*$, for all posterior $\mathrm{Q}$ being $\texttt{Poinc}(c_{\mathrm{P}})$ with $\mathrm{R}_\mathcal{D}(\mathrm{Q}) \leq C$ and such that for any $\mathbf{z} \in \mathcal{Z}$, $\ell(\cdot, \mathbf{z}) \in \mathrm{H}^1(\mathrm{Q})$:*

$$
\mathrm{R}_\mathcal{D}(\mathrm{Q}) \leq \frac{1}{1 - \frac{\lambda C}{2}} \left( \hat{\mathrm{R}}_{\mathcal{S}_m}(\mathrm{Q}) + \frac{\mathrm{KL}(\mathrm{Q},\mathrm{P}) + \log(1/\delta)}{\lambda m} \right)
$$
$$
+ \frac{\lambda}{2 - \lambda C} \left( c_P(\mathrm{Q}) \underset{\mathbf{z}\sim\mathcal{D}}{\mathbb{E}} \left[ \underset{h\sim\mathrm{Q}}{\mathbb{E}} \left( \|\nabla_h \ell(h,\mathbf{z})\|^2 \right) \right] + \underset{\mathbf{z}\sim\mathcal{D}}{\mathrm{Var}} \left( \underset{h\sim\mathrm{Q}}{\mathbb{E}}[\ell(h,\mathbf{z})] \right) \right).
$$

Theorem 22 highlights the influence of the gradient norm of $\nabla_h \ell(h, \mathbf{z})$ on the generalisation ability: small gradients make the bound vanish. The remaining variance term is not addressed at this stage and can be assumed to be bounded, but we cannot then recover a fast rate.

**Proof** We start from Chugg et al. (2023, Corollary 17), for any $\lambda > 0$, with probability at least $1 - \delta$, for all $m \in \mathbb{N}^*$, for all posteriors $\mathrm{Q} \in \mathcal{H}$, we have

$$
\mathrm{R}_\mathcal{D}(\mathrm{Q}) \leq \hat{\mathrm{R}}_{\mathcal{S}_m}(\mathrm{Q}) + \frac{\mathrm{KL}(\mathrm{Q},\mathrm{P}) + \log(1/\delta)}{\lambda m} + \frac{\lambda}{2} \left( \underset{h\sim\mathrm{Q}}{\mathbb{E}} \left[ \underset{\mathbf{z}\sim\mathcal{D}}{\mathbb{E}}[\ell(h,\mathbf{z})^2] \right] \right).
$$

The last term is then controlled as follows:

$$
\underset{h\sim\mathrm{Q}}{\mathbb{E}} \left[ \underset{\mathbf{z}\sim\mathcal{D}}{\mathbb{E}}[\ell(h,\mathbf{z})^2] \right] \leq \underset{\mathbf{z}\sim\mathcal{D}}{\mathbb{E}} \left[ c_P(\mathrm{Q}) \underset{h\sim\mathrm{Q}}{\mathbb{E}} \left( \|\nabla_h \ell(h,\mathbf{z})\|^2 \right) + \left( \underset{h\sim\mathrm{Q}}{\mathbb{E}}[\ell(h,\mathbf{z})] \right)^2 \right].
$$

We then introduce a supplementary variance term:

$$= \mathop{\mathbb{E}}_{\mathbf{z}\sim\mathcal{D}}\left[c_P(Q)\mathop{\mathbb{E}}_{h\sim Q}\left(\|\nabla_h\ell(h,\mathbf{z})\|^2\right)\right] + \mathop{\mathrm{Var}}_{\mathbf{z}\sim\mathcal{D}}\left(\mathop{\mathbb{E}}_{h\sim Q}[\ell(h,\mathbf{z})]\right)$$
$$+ \left(\mathop{\mathbb{E}}_{\mathbf{z}\sim\mathcal{D}}\mathop{\mathbb{E}}_{h\sim Q}[\ell(h,\mathbf{z})]\right)^2.$$

Note that by Fubini's theorem, the last term on the right-hand side is exactly $R_\mathcal{D}(Q)^2$. Then, using the fact that the averaged true risk is less than $C$, and reorganising the terms in Chugg et al. (2023, Corollary 17), we obtain, for $\lambda \in \left(0, \frac{2}{C}\right)$:

$$R_\mathcal{D}(Q) \le \frac{1}{1-\frac{\lambda C}{2}}\hat{R}_{\mathcal{S}_m}(Q) + \frac{\mathrm{KL}(Q,P)+\log(1/\delta)}{\lambda\left(1-\frac{\lambda C}{2}\right)m}$$
$$+ \frac{\lambda}{2-\lambda C}\left(c_P(Q)\mathop{\mathbb{E}}_{\mathbf{z}\sim\mathcal{D}}\left[\mathop{\mathbb{E}}_{h\sim Q}\left(\|\nabla_h\ell(h,\mathbf{z})\|^2\right)\right] + \mathop{\mathrm{Var}}_{\mathbf{z}\sim\mathcal{D}}\left(\mathop{\mathbb{E}}_{h\sim Q}[\ell(h,\mathbf{z})]\right)\right).$$

■

Now that Theorem 22 is proven, we only need to apply the Poincaré assumption on the data distribution to the variance term to derive Corollary 7.

### C.2. Proof of Theorem 8

**Theorem 8** *For any $C_1, C_2, c > 0$, for any data-free prior $P \in \mathcal{M}(\mathcal{H})$, for any loss function $\ell : \mathcal{H} \times \mathcal{Z} \to \mathbb{R}_+$ being $\mathcal{C}^2$ and for any $\delta \in (0,1]$, we have, with probability at least $1-\delta$ over the sample $\mathcal{S}$, for all $m \in \mathbb{N}^*$, for all $Q$ being $\texttt{Poinc}(c_P)=c$, $\texttt{QSB}(\ell, C_1)$, $\texttt{QSB}\left(\|\nabla_h\ell\|^2, C_2\right)$, and $\ell(\cdot,\mathbf{z}) \in \mathrm{H}^1(Q)$, and $\|\nabla_h\ell(\cdot,\mathbf{z})\|^2 \in \mathrm{H}^1(Q)$ for all $\mathbf{z}$,*

$$R_\mathcal{D}(Q) \le 2\hat{R}_{\mathcal{S}_m}(Q) + \frac{2c}{C_1}\mathop{\mathbb{E}}_{h\sim Q}\left[\frac{1}{m}\sum_{i=1}^m\|\nabla_h\ell(h,\mathbf{z}_i)\|^2\right]$$
$$+ 2\left(C_1 + c\frac{4cG^2+C_2}{C_1}\right)\frac{\mathrm{KL}(Q,P)+\log(2/\delta)}{m}.$$

**Proof** We start again from Theorem 6, with $\lambda = 1/C_1$, to obtain, with probability at least $1-\delta/2$:

$$R_\mathcal{D}(Q) \le 2\left(\hat{R}_{\mathcal{S}_m}(Q) + 2C_1\frac{\mathrm{KL}(Q,P)+\log(2/\delta)}{m}\right) + \frac{c_P(Q)}{C_1}\mathop{\mathbb{E}}_{\mathbf{z}\sim\mathcal{D}}\left[\mathop{\mathbb{E}}_{h\sim Q}\left(\|\nabla_h\ell(h,\mathbf{z})\|^2\right)\right]. \quad (4)$$

We now observe that $g : h, \mathbf{z} \mapsto \|\nabla_h\ell(h,\mathbf{z})\|^2$ is nonnegative. Given our assumptions, we apply the proof technique of Theorem 6 on $g$, *i.e.* we start again from Corollary 17 of (Chugg et al., 2023), apply Poincaré inequality on $Q$ and use the $\texttt{QSB}$ assumption on $g$. We then have, for any $\lambda > 0$, with probability at least $1-\delta/2$, for all $Q$ being $\texttt{Poinc}(c_P)$, $\texttt{QSB}(g, C_2)$ and $g(\cdot,\mathbf{z}) \in \mathrm{H}^1(Q)$ for

all $\mathbf{z} \in \mathcal{Z}$ :

$$
\mathbb{E}_{\mathbf{z} \sim \mathcal{D}} \left[ \mathbb{E}_{h \sim Q} \left( \|\nabla_h \ell(h, \mathbf{z})\|^2 \right) \right] \leq \mathbb{E}_{h \sim Q} \left[ \frac{1}{m} \sum_{i=1}^{m} \|\nabla_h \ell(h, \mathbf{z}_i)\|^2 \right] + \frac{\mathrm{KL}(Q, P) + \log(2/\delta)}{\lambda m}
$$
$$
+ \frac{\lambda c_P(Q)}{2} \mathbb{E}_{\mathbf{z} \sim \mathcal{D}} \left[ \mathbb{E}_{h \sim Q} \left( \|\nabla_h g(h, \mathbf{z})\|^2 \right) \right] + \frac{\lambda C_2}{2} \mathbb{E}_{\mathbf{z} \sim \mathcal{D}} \left[ \mathbb{E}_{h \sim Q} \left( \|\nabla_h \ell(h, \mathbf{z})\|^2 \right) \right]. \quad (5)
$$

Notice that by definition of $g(\cdot, \mathbf{z}) : \mathbb{R}^d \to \mathbb{R}$, we have $\nabla_h g(h, \mathbf{z}) = 2\mathrm{Hess}_h(\ell)(h, \mathbf{z})\nabla_h \ell(h, \mathbf{z})$, where $\mathrm{Hess}_h(\ell)$ denotes the Hessian of $\ell$. Thus, using the fact that $\ell(\cdot, \mathbf{z})$ is $G$ gradient-Lipschitz for any $\mathbf{z} \in \mathcal{Z}$, we get, for any $(h, \mathbf{z})$, that $\|\nabla_h g(h, \mathbf{z})\| \leq 2G\|\nabla_h \ell(h, \mathbf{z})\|$. Substituting this in Equation (5) gives:

$$
\mathbb{E}_{\mathbf{z} \sim \mathcal{D}} \left[ \mathbb{E}_{h \sim Q} \left( \|\nabla_h \ell(h, \mathbf{z})\|^2 \right) \right] \leq \mathbb{E}_{h \sim Q} \left[ \frac{1}{m} \sum_{i=1}^{m} \|\nabla_h \ell(h, \mathbf{z}_i)\|^2 \right] + \frac{\mathrm{KL}(Q, P) + \log(2/\delta)}{\lambda m}
$$
$$
+ \frac{\lambda}{2} \left( 4c_P(Q)G^2 + C_2 \right) \mathbb{E}_{\mathbf{z} \sim \mathcal{D}} \left[ \mathbb{E}_{h \sim Q} \left( \|\nabla_h \ell(h, \mathbf{z})\|^2 \right) \right]. \quad (6)
$$

Using that $c_P(Q) = c$, taking $\lambda = \frac{1}{4cG^2 + C_2}$ and reorganising the terms in Equation (6) gives:

$$
\mathbb{E}_{\mathbf{z} \sim \mathcal{D}} \left[ \mathbb{E}_{h \sim Q} \left( \|\nabla_h \ell(h, \mathbf{z})\|^2 \right) \right] \leq 2 \mathbb{E}_{h \sim Q} \left[ \frac{1}{m} \sum_{i=1}^{m} \|\nabla_h \ell(h, \mathbf{z}_i)\|^2 \right]
$$
$$
+ 2(4cG^2 + C_2)\frac{\mathrm{KL}(Q, P) + \log(2/\delta)}{m}. \quad (7)
$$

Finally, taking a union bound and plugging Equation (7) in Equation (4) concludes the proof. ∎

### C.3. Proof of Lemma 10

**Lemma 10** *For any $\gamma > 0$, for any $m \in \mathbb{N}^*$, for any data-free prior $P \in \mathcal{M}(\mathcal{H})$ being $L\text{-}Sob(c_{LS})$, for any loss function $\ell : \mathcal{H} \times \mathcal{Z} \to \mathbb{R}_+$ such that $\ell(\cdot, \mathbf{z}) \in \mathrm{H}^1(P)$ for any $\mathbf{z}$, we have*

$$
\mathrm{KL} \left( Q_{\mathcal{S}_m}^\gamma, P \right) \leq \frac{\gamma^2 c_{LS}(P)}{4} \mathbb{E}_{h \sim Q_{\mathcal{S}_m}^\gamma} \left[ \|\nabla_h \hat{R}_{\mathcal{S}_m}(h)\|^2 \right].
$$

**Proof** For conciseness, we rename $Q := Q_{\mathcal{S}_m}^\gamma$. We first note that we have

$$
\mathrm{KL} \left( Q_{\mathcal{S}_m}^\gamma, P \right) = \mathbb{E}_{h \sim Q} \left[ \log \left( \frac{dQ}{dP}(h) \right) \right]
$$
$$
= \underset{P}{\mathrm{Ent}} \left( \frac{dQ}{dP} \right) = \underset{P}{\mathrm{Ent}}[g^2],
$$

where $g = \sqrt{\frac{dQ}{dP}}$ and $\frac{dQ}{dP}$ is the Radon-Nikodym derivative of $Q$ with respect to $P$. Recall that $\frac{dQ}{dP}(h) = \frac{1}{Z} \exp(-\gamma \hat{R}_{\mathcal{S}_m}(h))$, where $Z = \mathbb{E}_{h \sim P}[\exp(-\gamma \hat{R}_{\mathcal{S}_m}(h))]$. Then, the function $g : h \mapsto$

$\frac{1}{\sqrt{Z}}\exp(-\frac{\gamma}{2}\hat{R}_{\mathcal{S}_m}(h))$ belongs to $\mathrm{H}^1(\mathrm{P})$ as long as $\ell \in \mathrm{H}^1(\mathrm{P})$. Indeed, since $\exp$ is infinitely smooth, $g \in \mathrm{D}_1(\mathbb{R}^d)$. Also, as the loss is nonnegative, we have $g \leq \frac{1}{\sqrt{Z}}$, so $g \in \mathrm{L}^2(\mathrm{P})$. Finally, we have $\nabla_h g(h) = -\frac{\gamma}{2}g(h)\nabla_h\hat{R}_{\mathcal{S}_m}(h)$. Since $g(h) \leq \frac{1}{\sqrt{K}}$, we only need to bound $\|\nabla_h\hat{R}_{\mathcal{S}_m}(h)\|^2$ to ensure that $g \in \mathrm{H}^1(\mathrm{P})$:

$$
\begin{aligned}
\|\nabla_h\hat{R}_{\mathcal{S}_m}(h)\|^2 &= \frac{1}{m^2}\sum_{1\leq i,j\leq m}\langle\nabla_h\ell(h,\mathbf{z}_i),\nabla_h\ell(h,\mathbf{z}_j)\rangle \\
&\leq \frac{1}{2m^2}\sum_{1\leq i,j\leq m}\|\nabla_h\ell(h,\mathbf{z}_i)\|^2 + \|\nabla_h\ell(h,\mathbf{z}_j)\|^2.
\end{aligned}
$$

Since we assumed $\|\nabla_h\ell(h,\mathbf{z})\|^2 \in \mathrm{L}^2(\mathrm{P})$ for all $\mathbf{z} \in \mathcal{Z}$, we conclude that $g \in \mathrm{H}^1(\mathrm{P})$. We then can apply the log-Sobolev inequality to conclude that

$$
\begin{aligned}
\mathrm{KL}\left(Q^\gamma_{\mathcal{S}_m},\mathrm{P}\right) &\leq c_{LS}(\mathrm{P})\underset{h\sim\mathrm{P}}{\mathbb{E}}[\|\nabla_h g(h)\|^2] \\
&= \frac{\gamma^2 c_{LS}(\mathrm{P})}{4}\underset{h\sim\mathrm{P}}{\mathbb{E}}\left[\|\nabla_h\hat{R}_{\mathcal{S}_m}(h)\|^2 g^2(h)\right] \\
&= \frac{\gamma^2 c_{LS}(\mathrm{P})}{4}\underset{h\sim\mathrm{P}}{\mathbb{E}}\left[\|\nabla_h\hat{R}_{\mathcal{S}_m}(h)\|^2\frac{d\mathrm{Q}}{d\mathrm{P}}(h)\right] \\
&= \frac{\gamma^2 c_{LS}(\mathrm{P})}{4}\underset{h\sim Q^\gamma_{\mathcal{S}_m}}{\mathbb{E}}\left[\|\nabla_h\hat{R}_{\mathcal{S}_m}(h)\|^2\right].
\end{aligned}
$$

∎

### C.4. Proof of Theorem 11

**Theorem 11** *We first have $C > 0$, $\gamma > 0$, and a data-free prior $\mathrm{P} \in \mathcal{M}(\mathcal{H})$ being $\mathtt{L\text{-}Sob}(c_{LS})$.*
*(i) For any loss function $\ell : \mathcal{H} \times \mathcal{Z} \to [0,1]$ and for any $\delta \in (0,1]$, with probability at least $1-\delta$ over the sample $\mathcal{S}$, for any $m \in \mathbb{N}^*$, we have*

$$
R_{\mathcal{D}}(Q^\gamma_{\mathcal{S}_m}) \leq 2\left(\hat{R}_{\mathcal{S}_m}(Q^\gamma_{\mathcal{S}_m}) + \frac{\gamma^2 c_{LS}(\mathrm{P})}{4m}\underset{h\sim Q^\gamma_{\mathcal{S}_m}}{\mathbb{E}}\left[\|\nabla_h\hat{R}_{\mathcal{S}_m}(h)\|^2\right] + \frac{\log(1/\delta)}{m}\right).
$$

*(ii) For any loss function $\ell$ and prior $\mathrm{P}$ satisfying the conditions of Proposition 4. Then, for any $\frac{2}{C} > \lambda > 0$, with probability at least $1-\delta$ over the sample $\mathcal{S}$, for all $m \in \mathbb{N}^*$, assuming $\ell(\cdot,\mathbf{z}) \in \mathrm{H}^1(Q^\gamma_{\mathcal{S}_m})$ and that $Q^\gamma_{\mathcal{S}_m}$ is $\mathtt{QSB}(C)$, we have*

$$
\begin{aligned}
R_{\mathcal{D}}(Q^\gamma_{\mathcal{S}_m}) \leq \frac{1}{1-\frac{\lambda C}{2}}&\left(\hat{R}_{\mathcal{S}_m}(Q^\gamma_{\mathcal{S}_m}) + \frac{\gamma^2 c_{LS}(\mathrm{P})}{4\lambda m}\underset{h\sim Q^\gamma_{\mathcal{S}_m}}{\mathbb{E}}\left[\|\nabla_h\hat{R}_{\mathcal{S}_m}(h)\|^2\right] + \frac{\log(1/\delta)}{\lambda m}\right. \\
&\left.+ \frac{\lambda e^{4\|\ell_2\|_\infty}c_{LS}(\mathrm{P})}{4-2\lambda C}\underset{\mathbf{z}\sim\mathcal{D}}{\mathbb{E}}\left[\underset{h\sim Q^\gamma_{\mathcal{S}_m}}{\mathbb{E}}\left(\|\nabla_h\ell(h,\mathbf{z})\|^2\right)\right]\right).
\end{aligned}
$$

**Proof** We start again from Chugg et al. (2023, Corollary 17), instantiated with a single $\lambda$. Then with probability at least $1 - \delta$, for all posteriors Q and for all $m \in \mathbb{N}^*$, we have

$$\mathrm{R}_{\mathcal{D}}(\mathrm{Q}) \leq \hat{\mathrm{R}}_{\mathcal{S}_m}(\mathrm{Q}) + \frac{\mathrm{KL}(\mathrm{Q}, \mathrm{P}) + \log(1/\delta)}{\lambda m} + \frac{\lambda}{2} \left( \underset{h \sim \mathrm{Q}}{\mathbb{E}} \left[ \underset{\mathbf{z} \sim \mathcal{D}}{\mathbb{E}} [\ell(h, \mathbf{z})^2] \right] \right).$$

For the first inequality, we simply take $\lambda = 1$, use the fact that $\ell(h, \mathbf{z})^2 \leq \ell(h, \mathbf{z})$, and reorganise the terms. Finally, we upper-bound the KL term using Lemma 10. For the second inequality, we apply Proposition 4 to use the fact that $\mathrm{Q}_{\mathcal{S}_m}^{\gamma}$ is $\mathtt{L\text{-}Sob}(c_{LS})$, and Proposition 15, which ensures that $\mathrm{Q}_{\mathcal{S}_m}^{\gamma}$ is $\mathtt{Poinc}(c_{\mathrm{P}})$, with constant equal to $c_{LS}\left(\mathrm{Q}_{\mathcal{S}_m}^{\gamma}\right)/2$. We then follow a proof technique similar to Theorem 6. We have :

$$\underset{h \sim \mathrm{Q}_{\mathcal{S}_m}^{\gamma}}{\mathbb{E}} \left[ \underset{\mathbf{z} \sim \mathcal{D}}{\mathbb{E}} [\ell(h, \mathbf{z})^2] \right] = \underset{\mathbf{z} \sim \mathcal{D}}{\mathbb{E}} \left[ \underset{h \sim \mathrm{Q}_{\mathcal{S}_m}^{\gamma}}{\mathrm{Var}} (\ell(h, \mathbf{z})) + \left( \underset{h \sim \mathrm{Q}_{\mathcal{S}_m}^{\gamma}}{\mathbb{E}} [\ell(h, \mathbf{z})] \right)^2 \right].$$

Applying Poincaré inequality then gives:

$$\leq \underset{\mathbf{z} \sim \mathcal{D}}{\mathbb{E}} \left[ \frac{c_{LS}(\mathrm{P})}{2} e^{4\|\ell_2\|_\infty} \underset{h \sim \mathrm{Q}_{\mathcal{S}_m}^{\gamma}}{\mathbb{E}} \left( \|\nabla_h \ell(h, \mathbf{z})\|^2 \right) + \left( \underset{h \sim \mathrm{Q}_{\mathcal{S}_m}^{\gamma}}{\mathbb{E}} [\ell(h, \mathbf{z})] \right)^2 \right].$$

Finally, using the fact that $\mathrm{Q}_{\mathcal{S}_m}^{\gamma}$ is $\mathtt{QSB}(\ell, C)$ allow us to reorganise the terms as in Theorem 6. Combining this with Lemma 10 to bound the KL divergence concludes the proof. ∎

### C.5. Proof of Theorem 12

**Theorem 12** *For any predictor set $\mathcal{H}$ with finite diameter $D > 0$, and for any function $f : \mathcal{H} \to \mathbb{R}$ satisfying $(\star)$, we have for all distributions $\mathrm{P} \in \mathcal{M}(\mathcal{H})$ and $\mathrm{Q} \in \mathcal{M}(\mathcal{H})$*

$$\underset{h \sim \mathrm{Q}}{\mathbb{E}}[f(h)] \leq \frac{G}{2} W_2^2(\mathrm{Q}, \mathrm{P}) + \underset{h \sim \mathrm{P}}{\mathbb{E}}[f(h)] + D \underset{h \sim \mathrm{Q}}{\mathbb{E}}[\|\nabla f(h)\|].$$

**Proof** We first assume that $G = 1$ in the $(\star)$ assumption. We start from the Kantorovich duality formula (Villani, 2009, Theorem 5.10), instantiated with the cost function $c(x, y) = \|x - y\|^2$. For any Q, P, since $W_2$ is a distance, we have:

$$W^2(\mathrm{Q}, \mathrm{P}) = W^2(\mathrm{P}, \mathrm{Q}) = \sup_{\phi, \psi} \underset{h \sim \mathrm{Q}}{\mathbb{E}}[\phi(h)] - \underset{h \sim \mathrm{P}}{\mathbb{E}}[\psi(h)], \tag{8}$$

where the supremum is taken over the functions $\phi, \psi \in L^1(\mathrm{Q}) \times L^1(\mathrm{P})$ such that for all $h, h' \in \mathcal{H}^2$, we have $\phi(h) - \psi(h') \leq \|h - h'\|^2$. We claim that if $\phi(h) = f(h) - D\|\nabla f(h)\|$ and $\psi(h') = f(h')$, then the pair $(\Phi, \Psi)$ satisfies $\phi(h) - \psi(h') \leq \frac{\|h - h'\|^2}{2}$. Indeed, we have

$$\phi(h) - \psi(h') = f(h) - f(h') - D\|\nabla f(h)\|$$
$$= f \circ g(1) - f \circ g(0) - D\|\nabla f(h)\|,$$

where $g(t) = th + (1-t)h'$. Then, by the fundamental theorem of calculus, we have

$$\phi(h) - \psi(h') = \int_0^1 (f \circ g)'(t)dt - D\|\nabla f(h)\|$$

$$= \int_0^1 \left\langle \nabla f\left(th + (1-t)h'\right), h - h'\right\rangle dt - D\|\nabla f(h)\|.$$

We now control the last term using that $\|h - h'\| \leq D$ and Cauchy-Schwarz inequality:

$$\phi(h) - \psi(h') \leq \int_0^1 \left\langle \nabla f\left(th + (1-t)h'\right), h - h'\right\rangle dt - \left\langle \nabla f(h), h - h'\right\rangle$$

$$= \int_0^1 \left\langle \nabla f\left(th + (1-t)h'\right) - \nabla f(h), h - h'\right\rangle dt.$$

Then by the $(\star)$ assumption:

$$\phi(h) - \psi(h') \leq \|h - h'\| \int_0^1 (1-t)dt \left\|h - h'\right\| dt$$

$$= \frac{\|h - h'\|^2}{2}.$$

We then conclude by applying Equation (8) to the pair $(2\phi, 2\psi)$. The general case with $G \neq 1$ follows immediately by considering the pair $(\frac{2}{G}\phi, \frac{2}{G}\psi)$. ∎

## C.6. Proof of Corollary 13

**Corollary 13** *For any distribution* P *being* L-Sob$(c_{LS})$ *such that* $dP(h) \propto \exp(-V(h))dh$*, with* $V$ *being* $\mathcal{C}^2$*, for any* $R > 0$*, for any function* $f$ *on the centred ball* $\mathcal{B}(\mathbf{0}, R)$ *of radius* $R$ *satisfying the* $(\star)$ *assumption, and for any distribution* $Q \in \mathcal{M}(\mathcal{H})$*, we have:*

$$\mathop{\mathbb{E}}_{h \sim Q}[f(\mathcal{P}_R(h))] \leq \frac{Gc_{LS}(P)}{4} \mathrm{KL}(Q, P) + \mathop{\mathbb{E}}_{h \sim P}[f(\mathcal{P}_R(h))] + 2R \mathop{\mathbb{E}}_{h \sim Q}[\|\nabla_h f(\mathcal{P}_R(h))\|],$$

*where* $\mathcal{P}_R$ *denotes the Euclidean projection onto* $\mathcal{B}(\mathbf{0}, R)$*.*

**Proof** We fix $R > 0$, and we start from Theorem 12 with predictor space $\mathcal{H}_0 = \mathcal{B}(\mathbf{0}, R)$, where $f$ is gradient-Lipschitz on this ball and the prior and the posterior are respectively $\mathcal{P}_R \# Q$ and $\mathcal{P}_R \# P$. We have

$$\mathop{\mathbb{E}}_{h \sim Q}[f(\mathcal{P}_R(h))] \leq \frac{G}{2}W_2^2(\mathcal{P}_R \# Q, \mathcal{P}_R \# P) + \mathop{\mathbb{E}}_{h \sim P}[f(\mathcal{P}_R(h))] + 2R \mathop{\mathbb{E}}_{h \sim Q}[\|\nabla_h f(\mathcal{P}_R(h))\|].$$

We first prove that $W_2^2(\mathcal{P}_R \# Q, \mathcal{P}_R \# P) \leq W_2^2(Q, P)$. Let $\pi \in \Gamma(Q, P)$ be the optimal transport coupling from P to Q, *i.e.*

$$W_2^2(Q, P) = \mathop{\mathbb{E}}_{(X,Y) \sim \pi}\left[\|X - Y\|^2\right].$$

Then, notice that if we denote by $\pi_1 = (\mathcal{P}_R, \mathcal{P}_R)\#\pi$, then $\pi_1 \in \Gamma(\mathcal{P}_R\#Q, \mathcal{P}_R\#P)$ and so:

$$
\begin{aligned}
W_2^2(\mathcal{P}_R\#Q, \mathcal{P}_R\#P) &\leq \mathop{\mathbb{E}}_{(X,Y)\sim\pi_1}\left[\|X-Y\|^2\right] \\
&= \mathop{\mathbb{E}}_{(X,Y)\sim\pi_1}\left[\|\mathcal{P}_R(X)-\mathcal{P}_R(Y)\|^2\right].
\end{aligned}
$$

Using the fact that $\mathcal{P}_R$ is 1-Lipschitz gives:

$$
\begin{aligned}
W_2^2(\mathcal{P}_R\#Q, \mathcal{P}_R\#P) &\leq \mathop{\mathbb{E}}_{(X,Y)\sim\pi_1}\left[\|X-Y\|^2\right] \\
&= W_2^2(Q, P).
\end{aligned}
$$

Next, we need to control $W_2^2(Q, P)$. To do so, we use the fact that P is $\texttt{L-Sob}(c_{LS})$ to assert, through Otto-Villani's theorem (Otto and Villani, 2000, Theorem 1) that the following holds: $W_2^2(Q, P) \leq \frac{c_{LS}(P)}{2}\,\mathrm{KL}(Q, P)$. This concludes the proof. ∎

### C.7. Proof of Theorem 14

**Theorem 14** *Let $\delta \in (0, 1)$ and $P \in \mathcal{M}(\mathcal{H})$ a data-free prior. Assume $\mathcal{H}$ has a finite diameter $D > 0$, for any loss function $\ell : \mathcal{H} \times \mathcal{Z} \to \mathbb{R}_+$ and any $m \in \mathbb{N}^*$, the generalisation gap $h \mapsto \mathrm{R}_{\mathcal{D}}(h) - \hat{\mathrm{R}}_{\mathcal{S}_m}(h)$ satisfies $(\star)$. Assume that $\mathbb{E}_{h\sim P}\,\mathbb{E}_{\mathbf{z}\sim\mathcal{D}}[\ell(h,\mathbf{z})^2] \leq \sigma^2$, then the following holds with probability at least $1 - \delta$, for any $m > 0$ and any $Q \in \mathcal{M}(\mathcal{H})$:*

$$
\mathrm{R}_{\mathcal{D}}(Q) \leq \hat{\mathrm{R}}_{\mathcal{S}_m}(Q) + \frac{G}{2}W_2^2(Q, P) + \sqrt{\frac{2\sigma^2\log\left(\frac{1}{\delta}\right)}{m}} + D\mathop{\mathbb{E}}_{h\sim Q}\left(\left\|\nabla_h\mathrm{R}_{\mathcal{D}}(h) - \nabla_h\hat{\mathrm{R}}_{\mathcal{S}_m}(h)\right\|\right).
$$

**Proof** We start from Theorem 12, using the fact that $\mathrm{R}_{\mathcal{D}}(h) - \hat{\mathrm{R}}_{\mathcal{S}_m}(h)$ is $G$-gradient-Lipschitz for any $m \in \mathbb{N}^*$ to obtain:

$$
\begin{aligned}
\mathop{\mathbb{E}}_{h\sim Q}[\mathrm{R}_{\mathcal{D}}(h) - \hat{\mathrm{R}}_{\mathcal{S}_m}(h)] \leq {}& \frac{G}{2}W_2^2(Q, P) + \mathop{\mathbb{E}}_{h\sim P}[\mathrm{R}_{\mathcal{D}}(h) - \hat{\mathrm{R}}_{\mathcal{S}_m}(h)] \\
&+ D\mathop{\mathbb{E}}_{h\sim Q}\left(\left\|\nabla_h\mathrm{R}_{\mathcal{D}}(h) - \nabla_h\hat{\mathrm{R}}_{\mathcal{S}_m}(h)\right\|\right)
\end{aligned}
$$

The only remaining term to control is $\mathbb{E}_{h\sim P}[\mathrm{R}_{\mathcal{D}}(Q) - \hat{\mathrm{R}}_{\mathcal{S}_m}(Q)]$. For this, we use the supermartingale concentration inequality of Chugg et al. (2023, Corollary 17) instantiated with the prior equals to the posterior, which shows that, for any $\lambda > 0$, with probability at least $1 - \delta$, we have

$$
\mathop{\mathbb{E}}_{h\sim P}[\mathrm{R}_{\mathcal{D}}(h) - \hat{\mathrm{R}}_{\mathcal{S}_m}(h)] \leq \frac{\log(1/\delta)}{\lambda} + \frac{\lambda}{2}\mathop{\mathbb{E}}_{h\sim P}\mathop{\mathbb{E}}_{\mathbf{z}\sim\mathcal{D}}[\ell(h,\mathbf{z})^2].
$$

The last term on the right-hand side is bounded by $\sigma^2$ by assumption. Taking $\lambda = \sqrt{\frac{2\log(1/\delta)}{\sigma^2}}$, we finally get $\mathbb{E}_{h\sim P}\,\mathbb{E}_{\mathbf{z}\sim\mathcal{D}}[\ell(h,\mathbf{z})^2] \leq \sqrt{2\log(1/\delta)/m}$, which concludes the proof. ∎

