# OpenReview forum: "A PAC-Bayesian Link Between Generalisation and Flat Minima"
_algorithmiclearningtheory.org/ALT/2025/Conference — ALT 2025_

### Official Review · Reviewer_tiMX · 2024-10-23
**Very exciting objective, but many mistakes and important questions not addressed**

**Rating:** 6
**Confidence:** 4

**Review:**

Summary:

There is empirical evidence that flat minima lead to better generalization in deep learning. This is often invoked to support Bayesian methods in machine learning. In particular, it was argued that flat minima will lead to tighter PAC-Bayes bounds [Dziugaite & Roy, 2017]. The objective of this paper is to go beyond heuristics, and to write PAC-Bayes bound that depend explicitly of the flatness of the minimum of the empirical risk. This is an excellent objective in my opinion. However, I'm not completely convinced by the outcome. After a very reasonable result (Theorem 6), the authors make some weird / approximative / misleading statements and loose space with unnecessary developments. On the other hand, they don't discuss key issues regarding how Theorem 6 can be useful in practice.

Major problems:

- page 5, Theorem 6: the result is nice and might lead to very exciting developments. But for the reader who wants to use these bounds, the first thing that should be discussed is: given a posterior $Q$, is it possible to check the conditions $Poinc(c_P)$ and $QSB(\ell,C)$?

- page 5: I find the discussion on "transitory fast-rates PAC-Bayes bounds" is misleading. The standard PAC-Bayes bound is in
$$ \sqrt{KL/m} \leq \inf_{u} \left[\frac{KL}{2u} + \frac{u}{2m}\right]. $$
Thus, by putting $u=2 m\epsilon$, we have
$$ \sqrt{KL/m} \leq \inf_{u} \frac{KL}{2\epsilon m} + \epsilon . $$
That is, all PAC-Bayes bounds are "transitory fast-rates". It's almost like a synonym for "standard bound". The difficulty for fast rates is actually to remove $\epsilon$. It is also weird that the authors cite [Alquier et al., 2016] as a reference for the $1/\sqrt{m}$ rate, which is simply the minimax rate (at least without additional assumptions on the loss) and can be found in all PAC-Bayes (and PAC) bounds.

- page 5: the discussion "high-probability bounds with fast rates, a paradox?" is in the same vein. First, I doubt that $\sigma$ is the variance of the dataset in Grunwald's statement. I think it is rather the variance of $\ell(h^*,z)$. And of course, it is known that if $\ell(h^*,z)=0$, convergence is faster than $1/\sqrt{m}$, so there is no contradition between fast rates and Grunwald's lower bounds. The whole discussion is useless anyway, as your "transitory fast-rates" bounds are actually "standard rates bounds" and do not contradict Grunwald's result.

- page 7: Theorem 8, there the empirical risk is multiplied by $2$. In the noiseless case, where $R_{\mathcal{S}m}(\hat{h})=0$, this is of course fine. But in the noisy case, this makes this bound very poor, because even when the sample size tends to infinity, you have no chance to recover the optimal prediction risk. It's OK to mutipliy $R_{\mathcal{S}m}(\hat{h})$ by a quantity larger than $1$, as long as this quantity tends to $1$ for large sample size. As it is stated, your result does even satisfy your definition of "transitory fast rates".

Frtextbfom the proof, this is due to the choice $\lambda=1/C_1$. It is necessary to allow $\lambda\rightarrow 0$ to get a non-trivial bound.

This problem is recurrent: Corollary 9 page 8, Theorem 11 (i) page 9.

- page 8: "... Gaussian distributions, but this class of distribution is restrictive. Following the approach of Catoni (2007), we go beyond the Gaussian distributions to focus on Gibbs posteriors". Gaussian distributions are feasible in practice, which is why [Dziugaite & Roy, 2017] used them for deep learning. Gibbs distributions might lead to feasible MCMC algorithms in simple models, but I doubt they are feasible in really deep networks. Of course, I believe it is interesting to study Gibbs posteriors as theoretical objects, but it's not fair to pretend that they are less limited than Gaussian. Actually, this leads to the question: can we apply Theorem 6 to Gaussian distributions, or are Gibbs distributions the only case where you can check the assumptions? This is one of the key that will make your results useful in practice. Thus, it should be discussed.

- page 10, Theorem 14: the term $W_2^2(Q,P)$ has no reason to go to zero, unless $Q\rightarrow P$? Or am I missing something? It seems to me you are adding a constant term to a PAC-Bayes bound, that makes it irrelevant.

Example: assume you have two classifiers $\mathcal{H}=\{h_1,h_2\}$, one of them is always right, say $h_1$, and the other one is always wrong, $h_2=1-h_1$. In practice we don't know which one is always right, so we take $P$ as uniform (a reasonable choice), and $Q$ is the point mass on the ERM, which will turn out to be $h_1$. Standard PAC-Bayes bound will give $R_{\mathcal{D}}(Q) \lesssim \sqrt{\log(2)/m} $. Fast rates-bounds will tell you $R_{\mathcal{D}}(Q) \lesssim \log(2)/m $. However, your bound will only guarantee $R_{\mathcal{D}}(Q) \lesssim constant = W_2^2(Q,P) = 1/2 d^2(h_1,h_2)$ where $d$ is the metric on your hypothesis space. This bound is not even able to prove the consistency of the ERM for 2 hypothesis in the noiseless case.

Is it really worth stating such a result?

- page 19: $(x_1+\dots+x_m)^2 \leq 2 (x_1^2 + \dots x_m^2)$ is wrong in general (take $x_1=\dots=x_m=1$ to see it). The best you can do is $(x_1+\dots+x_m)^2 \leq m (x_1^2 + \dots x_m^2)$. Thus, something in the proof of Corollary 19 seems wrong.

- page 20: "these bounds are relevant for practitioners as they require little computational time". Sampling from the Gibbs posterior is a computationaly difficult problem. Do you have a proof that you can sample from it using little computational time?

Minor comments:

- page 1: "there is no formal definition of flatness". The view in [Dziugaite & Roy, 2017] is that the flatness of the minimum at $\hat{h}$ can be measured by how large we can make the variance of a Gaussian $Q$ centered at $\hat{h}$ while keeping $R_{\mathcal{S}m}(\hat{h}) ~ E_{\sim Q}[R_{\mathcal{S}m}(h)]$. This can be turned into a quantitative definitions in a straightforward way, and it is a very practical notion. Catoni [Catoni, 2007], page 6-7, explicits a PAC-Bayes bound for classification in terms of the set $S=\{h: R_{\mathcal{S}m}(h)=R_{\mathcal{S}m}(\hat{h})\}$. When this set is large, the minimum is exactly flat on a large area, and the generalization bound depends on the mass of this set under the prior. For more general losses, $S$ can be replaced by $\{h: R_{\mathcal{S}m}(h) < R_{\mathcal{S}m}(\hat{h}) + \delta \}$ as in Def. 4.3 in [Alquier, 2024], it is a very classical condition that this set is large enough under the prior ("prior mass condition"). All these conditions are attempts to formalize flatness and should be discussed.

- page 4: Assumption 5 is not classical, so more discussion would be welcome. Can you check this assumtion in practice? Also, if you replace $\ell(h,z)$ by $\ell(h,z)-\ell(h^*,z)$ where $h^*$ is the optimal predictor, Assumption 5 is implied by the classical Bernstein condition, which also implies fast rates (bounds in 1/m instead of 1/sqrt(m)). Is Bernstein condition satisfied under Assumption 5? If so, all you results should be stated with fast-rates PAC-Bayes bounds.

- page 9, Theorem 11, (i): "and for all $Q\in\mathcal{H}$" seems useful as the following equation is not stated for any $Q$ but for the Gibbs posterior $Q_{\mathcal{S}_m}^\gamma$ only.

- page 9, Theorem 11, (ii): I appreciate the effort to keep the factor $1/(1-\lambda C/2)$ instead of $2$. However, the term $\|\ell_2\|_{\infty}$ in the bound is not defined (typo?).

Conclusion:

In my opinion, the authors should remove the discussions on "transitory fast-rate" bounds, the useless Wasserstein bound. They should instead discuss the application of their Theorem 6 beyond the theoretical objects (Gibbs distributions) to Gaussian distributions, or other feasible methods. With such results, this will be a nice contribution.

***********
Post-rebuttal update:

cf. the discussion below. All problems were fixed. If the authors can implement all the discussions mentioned in the discussion, I believe the paper is a reasonable contribution for COLT.

**Paper Award:**

No

---

> ### Author Response · Authors · 2024-11-22
> **Rebuttal pt 1**
>
> **Is it possible to check the conditions $QSB$ and $Poinc$?**
>
> Concerning the Poincaré condition, we will clarify in Propositions 3 and 4 that Gibbs posteriors with specific conditions and Gaussian distributions satisfy $Poinc$ with explicit constants. Furthermore, we will add below Corollary 7 that the work of (Schlichting, 2019) showed that mixtures of Gaussians also satisfy Poincaré's inequality (with an upper bound on the constant value); we will include this result below Proposition 4.
>
> Concerning the QSB condition, we provide an example below where the QSB condition is satisfied for an unbounded learning problem.
>     Indeed, assume the loss $\ell$ is a nonnegative function that is $L$-Lipschitz w.r.t. $h\in\mathcal{H}$.
> Moreover, assume that we are in the realisable case, which means that  there exists $h^*\in\mathcal{H}$ such that $\forall \mathbf{z}\in\mathcal{Z}, \ell(h^*,\mathbf{z})=0$, and that $Q$ is an arbitrary distribution such that the mean $m\_Q$ and the standard deviation $\sigma_Q$ are both bounded by a certain $K$. Then, by Lipschitzness and Cauchy–Schwarz's inequality, for any $\mathbf{z}\in\mathcal{Z}$, we have, because $\ell(h,\mathbf{z})= \ell(h,\mathbf{z})-\ell(h^*,\mathbf{z})$, $\mathbb{E}\_{Q}[\ell(h,\mathbf{z})]^2 \leq \left(\mathbb{E}\_{Q}[\sqrt{L \|h-h^*\|\ell(h,\mathbf{z})}]\right)^2 \leq L\mathbb{E}\_{Q}[\|h-h^*\|] \mathbb{E}\_{Q}[\ell(h,\mathbf{z})]$.
> Finally, note that by Jensen and a bias-variance decomposition, we have
> $\mathbb{E}\_{Q}[\|h-h^*\|] \leq \sqrt{\mathbb{E}\_{Q}[\|h-h^*\|^2]} = \sqrt{\sigma_Q^2 +\|m\_Q - h^*\|^2}\leq \sqrt{K^2 + (K+\|h^*\|)^2}$.
> This ensures that the QSB condition holds in this case with constant $C= L\sqrt{K^2 + (K+\|h^*\|)^2}$.
>
> **About transitory fast rates**
>
> We thank you for your remark. The main difference between transitory fast rates and classical PAC-Bayes bounds is that transitory fast rates hold \textbf{with probability at least $1-\delta$, for all $m\in\mathbb{N}^*$}, while classical bounds hold \textbf{for all $m\in\mathbb{N}^*$, with probability at least $1-\delta$}.
> To understand the difference between these two properties, we first abandon the name transitory fast rate and redefine the concept as follows.
> We now refer to *time uniform estimation* PAC-Bayes bounds, i.e., bounds such that there exists $\mathcal{C} \subseteq \mathcal{M}(\mathcal{H})$, a positive threshold sequence $(\varepsilon\_m)_{m\in\mathbb{N}^*}$, and $\alpha>0$ with probability at least $1-\delta$, for all $Q\in \mathcal{C}$, and $m\in\mathbb{N}^*$ we have
> $R\_{\mathcal{D}}(Q) \leq \frac{\alpha}{m}\left[KL(Q,P) +\log(1/\delta)\right] + \varepsilon\_m.$
> Theorem 6 satisfies this definition with $\alpha= \frac{2}{\lambda(2-\lambda C)}$ and $\varepsilon\_m = \frac{2}{2-\lambda C}\left(\hat{R}\_{\mathcal{S}\_m}(Q) + \frac{\lambda}{2} \mathbb{E}\_{\mathcal{D},Q}\left[\|\nabla_h \ell(h,\mathbf{z})\|^2\right]\right)$.
>
> In particular, if there exists an upper bound on $\sup_{m\geq m_0} \varepsilon_m$, then we retrieve the notion of transitory fast rates as proposed in our paper, which holds simultaneously for all $m\geq m_0$. Furthermore, by setting $\lambda= 1/C\sqrt{m}$ in Theorem 6, the time-uniform property is lost, and we obtain a slow convergence rate for a single $m$, similar to classical PAC-Bayes bounds.
> Now we claim that the fundamental difference between classical PAC-Bayes and time-uniform estimation bounds is that the former allows in probability convergence while the latter allows almost surely convergence.
> Indeed, let us first consider a nonnegative loss $\ell$, a prior $P$, a countable dataset $\mathcal{S}$, and assume there exists a sequence $(Q\_m)_{m\in\mathbb{N}^*}$ which is such that $KL(Q\_m,P)\leq D$ for all $m$ and both $\hat{R}\_{\mathcal{S}\_m}(Q\_m)$ and $\mathbb{E}\_{\mathbf{z}\sim\mathcal{D},h\sim Q\_m}\left[\|\nabla\_h \ell(h,\mathbf{z})\|^2\right]$ goes to zero as $m$ goes to infinity almost surely.
>
> Then, the classical McAllester's bound states that for any $m\in\mathbb{N}^*$, and $\delta\_m\in(0,1]$, with probability at least $1-\delta\_m$, we have $\mathbb{P}\left( R_{\mathcal{D}}(Q\_m)   \leq R_{\mathcal{S}\_m}(Q\_m) + \sqrt{\frac{KL(Q\_m,P) +\log(2\sqrt{m}/\delta\_m)}{2m}}\right) \geq 1- \delta\_m$.
> Let $\alpha>0$ and take for all $m\in\mathbb{N}^*$, the confidence parameter $\delta\_m= \delta/m$. Since we assume that $R\_{\mathcal{S}\_m}(Q\_m) \rightarrow 0$, the square root in McAllester's bound goes to zero. Thus, there exists $m\_0\in\mathbb{N}^*$ such that for all $m\geq m\_0$, we have  $R\_{\mathcal{D}}(Q\_m)   \leq R\_{\mathcal{S}\_m}(Q\_m) + \sqrt{\frac{KL(Q\_m,P) +\log(2m\sqrt{m}/\delta)}{2m}}  \leq \alpha$. Thus, for all $m \geq m\_0$, we know that $\mathbb{P}\left(R\_{\mathcal{D}}(Q\_m) < \alpha  \right) \geq 1-\delta/m$.
> Taking the limit as $m$ goes to infinity ensures that for any $\alpha>0$, we have $\lim\_{m\rightarrow \infty} \mathbb{P}\left(R\_{\mathcal{D}}(Q\_m) < \alpha  \right) = 1$.

---

> > ### Author Response · Authors · 2024-11-22
> > **Rebuttal pt 2**
> >
> > Thus, the classical McAllester's bound allows the sequence $R_{\mathcal{D}}(Q_m)$ to converge in probability to zero.
> >
> > In contrast, our Theorem 6 with $\lambda = 1/C$ is a time uniform estimation bound with $\alpha= 2C$ and $\varepsilon\_m = 2\left(\hat{R}\_{\mathcal{S}\_m}(Q) + \frac{1}{2C} \mathbb{E}\_{\mathcal{D},Q}\left[\|\nabla\_h \ell(h,\mathbf{z})\|^2\right]\right)$.
> > Then, thanks to our assumptions, we know that both $\frac{\alpha}{m}\left[KL(Q,P) +\log(1/\delta)\right]$ and $\varepsilon\_m$ goes to zero as $m$ goes to infinity, we can then take the limit: for any $\delta>0$, we have $\mathbb{P}\left(\lim\_{m\rightarrow\infty}R\_{\mathcal{D}}(Q\_m) =0  \right) \geq 1-\delta$.
> > As $\lim\_{m\rightarrow\infty}R\_{\mathcal{D}}(Q\_m)$ does not depend on $\delta$, we can then make $\delta$ go to zero to obtain $\mathbb{P}\left(\lim\_{m\rightarrow\infty}R\_{\mathcal{D}}(Q\_m) =0  \right)=1$.
> > Thus, $R\_{\mathcal{D}}(Q\_m)$ converges almost surely to zero.
> > Such a conclusion cannot be achieved with classical PAC-Bayes bounds; our bound is stronger as it allows an almost sure convergence, which cannot be obtained by simple manipulations of classical bounds.
> >
> > **Discussion "high-probability bounds with fast rates, a paradox?"**
> >
> > While it is true that the variance is the one of the loss $\ell(h,\mathbf{z})$ and not the one of the dataset directly; we clarified it in the paper.
> > Moreover, the example of Grünwald et al. is valid only for a singleton $\mathcal{H}=\{h\}$ and does not imply the existence of $h^*$ such that for all $\mathbf{z}\in\mathcal{Z}$, we have $\ell(h^*,\mathbf{z})=0$. Such an assumption is challenging and may not be satisfied for complex structures (e.g. deep neural nets). Thus, we believe that this discussion provides valuable context. We thank you for your comment, which will help us clarify this point.
> >
> > **About Theorem 8**
> >
> > Our Theorem 8 can be extended to include two parameters, $\lambda_1$ and $\lambda_2$, which would yield a bound with a shape similar to that of Theorem 6. As stated in the first version of our paper, our result is compatible with the notion of a time-uniform estimation bound and is presented in an accessible manner. However, we will include a complete version with those parameters in the appendix.
> >
> > **About restrictiveness of Gaussian measures**
> >
> > We believe there is a misunderstanding. We do not suggest that Gaussian measures are restrictive, as they are among the best choices for practical applications due to the closed form of the KL divergence. Our point is that bounds, where the KL divergence is controllable only for Gaussian distributions, are restrictive. The goal of this section is to provide bounds that allow controlling the KL divergence for Gibbs posteriors. We will correct this.
> >
> > **About the Wasserstein bound**
> >
> > We thank you for raising this.
> > Our Wasserstein PAC-Bayes bound may have a hidden convergence rate in the Lipschitz constant of the gradient. This is the case in your example where $\mathcal{H}= \{h\_1,h\_2\}$.
> >
> > Indeed, it is important to note that in Theorem 12, the condition '$f$ has $G$-Lipschitz gradients' can be relaxed.
> > Specifically, $\forall (a,a')\in\mathcal{H}$, the term $\langle \nabla f,a-a'\rangle$ is $G\|a-a'\|$-Lipschitz w.r.t. the hypotheses in $\mathcal{H}$.
> > This relaxation is valid because the entire proof relies on the fundamental theorem of calculus.
> >
> > Then, consider Theorem 14, we assume that the loss $\langle\nabla\_h\ell(\cdot,\mathbf{z}),a-a'\rangle$ is $G\|a-a'\|$-Lipschitz for all $\mathbf{z}\in\mathcal{Z}$ and $a,a'\in \mathcal{H}^2$ (weaker than assuming $\ell(\cdot,\mathbf{z})$ is $G$-gradient Lipschitz for all $\mathbf{z}\in\mathcal{Z}$). Then with probability at least $1-\delta$, the generalisation gap $\Delta\_{\mathcal{S}\_m} = R\_{\mathcal{D}} - \hat{R}\_{\mathcal{S}_m}$ satisfies for $a,a'\in \mathcal{H}^2$, $\langle \nabla \Delta\_{\mathcal{S}\_m},a-a'\rangle$ is $G\|a-a'\|\sqrt{\frac{2\log(4/\delta)}{m}}$- gradient-Lipschitz over $\mathcal{H}$.
> > Indeed, fix $a,a'\in\mathcal{H}^2$, then $$\left\langle\nabla\Delta\_{\mathcal{S}\_m}(h\_1)-\nabla\Delta\_{\mathcal{S}\_m}(h\_2), a-a' \right\rangle = \frac{1}{m} \sum\_{i=1}^m  \left\langle\nabla\ell(h\_2,\mathbf{z}\_i) - \nabla\ell(h\_1,\mathbf{z}\_i) -(\nabla R\_{\mathcal{D}}(h\_2) -\nabla R\_{\mathcal{D}}(h\_1)),a -a'\right\rangle.$$
> > This is a sum of i.i.d. random variables with mean $0$.
> > Moreover, by Lipschitzness of $\langle\nabla \ell, a\rangle$ for all $\mathbf{z}\in\mathcal{Z}$, we know that $|\langle \nabla\ell(h\_2,\mathbf{z}) - \nabla\ell(h\_1,\mathbf{z}),a - a' \rangle| \leq G\|a-a'\|\cdot\|h\_1-h\_2\|$.
> > Then, by Hoeffding's inequality, with probability $1-\delta/2$, we have $\left|\left\langle\nabla\Delta\_{\mathcal{S}\_m}(h\_1)-\nabla\Delta\_{\mathcal{S}\_m}(h_2), a - a' \right\rangle \right| \leq G\|a-a'\|\cdot\|h_1-h_2\|\sqrt{\frac{2\log(4/\delta)}{m}}$.
> > Taking a union bound on the value of $(a,a') = (h\_1,h\_2)$ or $(h\_2,h\_1)$ concludes the proof.

---

> > ### Author Response · Authors · 2024-11-22
> > **Rebuttal pt 3**
> >
> > Then, this proves that in your example, we are able to recover a convergence rate. The idea of this proof is not new and can be traced back to (Amit et al., 2022) for Lipschitz losses and 1-Wasserstein distance. This proof technique yields a convergence rate for any finite $\mathcal{H}$ (but you have to pay a $\log(|\mathcal{H}|^4)$ factor due to union bounds.
> >
> > We will add this general corollary and update the underlying condition of Theorem 12 in the next version, thank you for this discussion.
> >
> > **Pages 19 and 20**
> >
> > For the remark on page 19, you are right, the true argument is as follows: as the loss is $L$-Lipschitz w.r.t. $\mathbf{z}\in\mathcal{Z}$, for any dataset $\mathcal{S}\_m = (\mathbf{z}\_1\cdots,\mathbf{z}\_m)$, any $\mathcal{S}'\_m = (\mathbf{z}'\_1\cdots,\mathbf{z}'\_m)$, and any $h\in \mathcal{H}$, we have
> > $\Delta_{\mathcal{S}\_m}(h) - \Delta\_{\mathcal{S}'\_m}(h) \leq \frac{L}{m} \sum\_{i=1}^m \|\mathbf{z}\_i-\mathbf{z}'\_i\|= \frac{L}{m} \sum_{i=1}^m \sqrt{\|\mathbf{z}_i-\mathbf{z}'_i\|^2}$.
> > Then, by the concavity of the square root, we have
> > $\Delta\_{\mathcal{S}\_m}(h) - \Delta\_{\mathcal{S}'_m}(h) \leq L\sqrt{\frac{1}{m}\sum\_{i=1}^m \|\mathbf{z}\_i-\mathbf{z}'\_i\|^2} = L\frac{\|\mathcal{S}\_m - \mathcal{S}'\_m\|}{\sqrt{m}}$.
> > The underlying norm $\|\mathcal{S}'\_m\|$ is the one derived from the scalar product $\langle \mathcal{S}\_m, \mathcal{S}'\_m\rangle= \sum\_{i=1}^m \langle \mathbf{z}\_i,\mathbf{z}'\_i\rangle$.
> >
> >
> > Concerning page 20, our point is that disintegrated PAC-Bayes bounds hold for a single draw of the posterior. Thus, if you draw, for instance, from Gaussian distributions, you do not need to draw many points to perform a Monte Carlo approximation of $\mathbb{E}_{Q}(\hat{R}_{\mathcal{S}_m}(h)]$.
> > A similar reasoning applies to Gibbs posteriors, you only need one draw (which could be costly, but it is better to have 1 draw than 500 samples) to have a valid bound.
> >
> > **Minor comments**
> >
> > - Thank you for this. We should have stated, 'there is no globally accepted definition of flatness' as there are many, possibly incompatible, definitions of flatness in the literature. We will add a discussion on this point.
> >
> > - We have already discussed the difference between PAC-Bayes Bernstein bounds and ours in section A.3.
> > A short summary of this section is that PAC-Bayes Bernstein bounds mixed with the Poincaré inequality are linked to the gradient $\nabla_\mathbf{z} \ell(h,\mathbf{z})$ while the QSB is different, being linked to the gradient $\nabla_h\ell(h,\mathbf{z})$ which, to our knowledge, is novel in PAC-Bayes. In the section 'Is it possible to check Poinc and QSB?', we provided a concrete example where the QSB is satisfied for an unbounded Lipschitz loss.
> >
> > - Thank you, the sentence 'for all $Q$' is a typo here.
> >
> > - This a typo; thank you. We will correct it.
> >
> > Finally, we would like to thank you again for your insightful review and are happy to answer any additional questions you may have.

---

> > > ### Comment · Reviewer_tiMX · 2024-11-22
> > > **1) Thanks 2) Many problems solved 3) 2 problems remain**
> > >
> > > Thanks. I sincerely recognize the authors addressed many of the problems I raised seriously. Two problems remain that are quite serious in my opinion, and I'm reluctant to recommend to accept the paper unless they are completely solved.
> > >
> > > ** Is it possible to check the conditions QSB and Poinc?
> > >
> > > Thanks. Problem solved.
> > >
> > > ** About transitory fast rates
> > >
> > > I missed the "uniform with respect to m" aspect, which I acknoweldge is of interest. However, the discussion in the paper was very misleading, because the 1/m vs. 1/sqrt(m) is a different issue. I appreciate your suggestion to rephrase the whole to make it clearer.
> > >
> > > ** Discussion "high-probability bounds with fast rates, a paradox?"
> > >
> > > I maintain that the so-called paradox discussed in this paragraph is classical and well understood, and that the notion of "transitory fast rates" was not helfpul in this discussion.
> > >
> > > ** About Theorem 8
> > >
> > > Thank you. Whatever the reason, to multiply the empirical risk by a factor that will not converge to 1 makes the result extremely weak to the point that it is not interesting. Thus, as you suggest, it should be stated with lambda_1 and lambda_2, even if it makes the result less simple to read: it's the only way to keep it not trivial.
> > >
> > > ** About restrictiveness of Gaussian measures
> > >
> > > Thank. This looks okay to me.
> > >
> > > ** About the Wasserstein bound
> > >
> > > [PROBLEM NOT SOLVED]. Maybe I missed something, but I don't understand as the derivations you provided invalidate my claim. Theorem 14 as stated is completely useless, because the W2(P,Q) term will prevent the bound from converging to 0. In the derivations you provided, where is this term?? For now, I maintain: I can't see how this bound can converge to 0 in any case at all.
> > >
> > > Even if the bound could converge to 0 (I can't really see how, but I might be missing something), one question remains: why is Theorem 14 relevant? If the only example you can make it work on is the finite case, it's really weak. We can do the same with KL-PAC-Bayes. Should I understand that Wasserstein-PAC-Bayes only application is to mimic KL-PAC-Bayes in the finite case? Do the authors really believe it is worth stating such a result?
> > >
> > > I'm not against replacing KL by Wasserstein or anything else, but something must be achieved by doing so. There's no point in doing it for the sake of doing it.
> > >
> > > You can try to convince me that the bound is useful if you really think it is, but note that if you simply remove Section 5, I would consider the problem as solved.
> > >
> > > ** Pages 19 and 20
> > >
> > > Page 19: Thank you for the explanations. I believe this is now correct.
> > >
> > > Page 20: [PROBLEM NOT SOLVED]. I was not referring to the Gaussian case, but to the general Gibbs posterior. Once again, I don't think it's honnest to write that it's easy to sample from a Gibbs posterior if you are not able to provide an algorithm that do so provably. It has nothing to do with producing a single sample rather than many. If you are thinking about MCMC: once you reach the stationary regime, it becomes easy to get multiple samples anyway. The question is: how long should be the burnin period? This is a very difficult problem, and if the risk is a complex enough function, the burnin period can be arbitrarily large.
> > >
> > > I will not accept the claim "it's easy to sample from the Gibbs posterior", unless you provide explicitly an algorithm that do so (in a reasonable time) provably. It's NOT the case.
> > >
> > > There is a whole part of the literature about sampling provably from Gibbs posteriors. See among many others Durmus, Brosse & Moulines (2017) "Sampling from a log-concave distribution with compact support with proximal Langevin Monte Carlo". But their results rely on assumptions (log-concave distributions) that are certainly not satisfied in the examples you have in mind (neural networks). Your current claim is both dishonest, and dismissive of this whole direction of research.
> > >
> > > ** Minor comments
> > >
> > > Thanks, problem(s) solved.

---

> > ### Author Response · Authors · 2024-11-23
> > **Thank you for your answer, some clarifications below.**
> >
> > Thank you for your quick reply, we are happy to read that most of your concern are addressed, we answer your two remaining concerns below.
> >
> > **About Wasserstein bound**
> >
> > First, let us show you the relevance of involving Wasserstein distances in PAC-Bayes is significant. A major limitation of KL-based PAC-Bayes bounds is that Dirac distribution cannot be directly considered, although they appear in a lot of situations in machine learning. Wasserstein PAC-Bayes learning, which has recently been investigated in [1,2,3] allow Diracs. Precisely, the initial 1-Wasserstein PAC-Bayes bound of [1, Theorem 11] has no explicit convergence rate to attenuate the $1$-Wasserstein, while not being limited to the finite case (which they investigate in their Theorem 12). Our Theorem 14 ihas a similar shape than [1, Theorem 11]. Although this apparent weakness, the work of [2, Theorem 8] managed to get an explicit convergence rate under the 1-Wasserstein for infinite predictors spaces at the cost of the dimension. Investigating the toolbox of [2] to extend Theorem 14 is a promising future lead.
> > Moreover, even without explicit convergence rate under the Wasserstein, Wasserstein PAC-Bayes bound are useful in practice as they can be transformed into learning algorithms with good performances. In [3], authors created PAC-Bayes algorithms with the Wasserstein distance, allowing for the first time to directly consider Dirac distributions for both batch and online PAC-Bayes algorithms and showed the empirical efficiency of their methods. However, a limitation of all those works is that they require a lipschitz loss. Here, we consider an alternative assumption, satisfied with gradient-lipschitz losses. Having gradient-lipschitz losses is important in optimisation as it is often an alternative to convexity to obtain convergence guarantees for gradient descent and variants [5]. For all these reasons, we believe it is important to present Theorem 12 as it is a first step towards exciting practical and theoretical extensions.
> >
> > **Finite hypothesis case**
> > Now, let use get back to your example where $\mathcal{H}=\{h\_1, h\_2\}$. In our first answer, we showed that, for all $(a,a')\in \mathcal{H}^2, \langle \nabla \Delta_{\mathcal{S}_m},a-a'\rangle$ is $G\|a-a'\|\sqrt{\frac{2\log(8/\delta)}{m}}$-gradient-Lipschitz over $\mathcal{H}$ with probability at least $1-\delta/2$, which is the underlying condition to have Theorem 12 and Theorem 14 (as noticed in our first answer). Hence, at the cost of a union bound, we can apply Theorem 14 with the constant $G'= G\sqrt{\frac{2\log(8/\delta)}{m}}$ to obtain the following bound that holds with probability at least $1-\delta$.
> >
> > _Theorem 14 (Finite case)._ Let $\delta\in(0,1)$ and $P\in\mathcal{M}(\mathcal{H})$ a data-free prior.
> > Assume $\mathcal{H}$ is finite and has a finite diameter $D>0$, $\ell\geq 0$ is $G$-gradient lipschitz.  Assume $\mathbb{E}\_{h\sim P}\mathbb{E}\_{z\sim\mathcal{D}}[\ell(h,z)^2] \leq \sigma^2$, then the following holds, for any $m>0$, with probability at least $1-\delta$, for any $Q$:
> >
> > $R\_D(Q) \leq \hat{R}\_{\mathcal{S}\_m}(Q) + G\sqrt{\frac{\log(8/\delta)}{2m}} W\_2^2(Q,P) + \sqrt{\frac{2\sigma^2\log\left( \frac{4}{\delta} \right)}{m}} + D \mathbb{E}_{h\sim Q}\left( \left\| \nabla\_h R\_{\mathcal{D}}(h) - \nabla\_h \hat{R}\_{\mathcal{S}_m}(h) \right\| \right)$.
> >
> > Put into words, this shows that we can recover a convergence rate for the Wasserstein case in your example, which can also be extended to any finite $\mathcal{H}$ (at the cost of a $\log(|\mathcal{H}|^4)$ factor due to union bounds). We also suspect that this is a step forward a more general case where the bound holds for any set $\mathcal{H}$ as in [4].
> >
> > **page 20**
> >
> > We apologize for not realizing that the sentence, 'These bounds are relevant for practitioners as they require little computational time.' could be unclear. In fact, we completely agree that sampling from the Gibbs distribution is not an easy task, and we are sorry if it can be understood this way. Hence, we propose modifying the sentence to: 'These bounds may be relevant for practitioners when sampling is easy, as in the case of Gaussian distributions, since they require little computational time.'.
> >
> >
> > Again, we thank you for your time and your remarks, which both help a lot to improve our work.
> >
> >
> > **References**
> >
> > [1] Ron Amit, Baruch Epstein, Shay Moran, Ron Meir. Integral Probability Metrics PAC-Bayes Bounds, NeurIPS 2022
> >
> > [2] Maxime Haddouche, Benjamin Guedj. Wasserstein PAC-Bayes Learning: A Bridge Between Generalisation and Optimisation. CoRR. 2023
> >
> > [3] Paul Viallard, Maxime Haddouche, Umut Simsekli, and Benjamin Guedj. Learning via Wasserstein-Based High Probability Generalisation Bounds. NeurIPS. 2023
> >
> > [4] Paul Viallard, Maxime Haddouche, Umut Simsekli, and Benjamin Guedj. Tighter Generalisation Bounds via Interpolation. CoRR. 2024
> >
> > [5] G. Garrigos, R. M. Gower. Handbook of Convergence Theorems for (Stochastic) Gradient Methods. CoRR. 2023

---

> > > ### Comment · Reviewer_tiMX · 2024-11-24
> > > **OK!**
> > >
> > > *********
> > >
> > > Page 20: problem solved, thanks.
> > >
> > > *********
> > >
> > > Wasserstein: 1) thanks for your explanations. I understand now that this bound can lead to non-trivial results. Sorry that I misunderstood your explanations the first time, everything was in your answer, my mistake. 2) I'm still no impressed by Theorem 14, in the sense that I can't really imagine a single situation where it would improve on standard KL-PAC-Bayes. I'm not saying it is not the case, I'm just saying that it's not proven yet.
> > >
> > > Indeed:
> > >
> > > The argument "Wasserstein allows Dirac posteriors" is weak. Indeed, with KL-PAC-Bayes, it's quite standard to "approximate" Dirac masses by uniform distributions on very small balls (for example Catoni etc.). Of course, Wasserstein seems simpler in the sense that you can remove this approximation step, but in the end, this is only useful if the rate is at least as good. Once again, I would like to see the bound in action in a non-trivial example together with a proof that the bound is not worse than the KL one in this example.
> > >
> > > Note that the "finite H case" (which I introduced in the discussion for my counterexample) is also not particularly convincing in defense of Wasserstein-PAC-Bayes: in this case, KL-PAC-Bayes naturally allows a Dirac mass posterior.
> > >
> > > That being said, I totally acknowledge that this bound leads to non-trivial results. It might totally be the case that it turns out to be very useful in the future. But I hope the authors will understand it's also a little puzzling to throw new bounds without taking the time to explore the settings where there would be useful. I STRONGLY ENCOURAGE THEM TO DO THIS WORK IN THE FUTURE.
> > >
> > > In the meantime, I thank the authors for this thorough and fruitful discussion. I will increase my score (assuming the authors will include in the paper the discussion on the Lipschitz constants G and G' to explain to the readers that Theorem 14 can lead to rates of convergence).

---

> > > > ### Author Response · Authors · 2024-11-24
> > > > **Thank you!**
> > > >
> > > > We agree that there is still exciting leads in developing Theorem 14 to reach stronger results and that we should have at least, discuss more about the usefulness of Wasserstein PAC-Bayes bounds and about the interest of introducing the 2-Wasserstein before stating Theorems 12-14, we will add all the discussion written here to make this clearer.
> > > > In the meantime we will, of course, add a novel corollary in our work to tackle the case of finite H to show Theorem 14 can lead to non-trivial bounds.
> > > >
> > > > More generally, we will add all the discussion we had here in the revised version of our work, they are important.
> > > >
> > > > Thank you for having this discussion with us, it has been a pleasure.

---

> > > > > ### Author Response · Authors · 2024-11-25
> > > > >
> > > > > Dear reviewer,
> > > > >
> > > > > Thank you once again your constructive feedback. We noticed that your comments indicate a positive assessment of our work. Would you consider updating your score to reflect this? As the rebuttal period will end soon, we wish to have the final scores for the decision period.
> > > > >
> > > > > We greatly appreciate your time and effort.
> > > > >
> > > > > Best regards,
> > > > > Authors

---

> > > > > > ### Comment · Reviewer_tiMX · 2024-11-25
> > > > > >
> > > > > > Dear authors,
> > > > > > Indeed, I mentioned I will increase my score. I tried, but for now, OpenReview does not allow me to edit my review nor my scores. I assume this will be allowed at the end of the discussion period, before the decisions.
> > > > > > In any case, I will let the AC know.

---

> > > > > > > ### Author Response · Authors · 2024-11-25
> > > > > > >
> > > > > > > Thank you very much!
> > > > > > >
> > > > > > > Best,
> > > > > > > Authors

---

### Official Review · Reviewer_sSsB · 2024-11-05
**Has important result regarding flatness but seemed to have overlooked previous fast 1/m rate.**

**Rating:** 7
**Confidence:** 3

**Review:**

The authors proved results of PAC bounds by keeping the gradients of the loss function in mind. In particular, given that the of the loss functions have ''well-formed'' gradients (i.e. obeying Sobolev inequality)
	and has gradient decaying to 0 near the origin (flatness),
	this PAC bound could potentially give a ''fast'' $\frac{1}{m}$ rate (in a ''flat'' scenario), as opposed to the ''slow'' $\sqrt{\frac{1}{m}}$ rate (typically achieved by generalists).
	This seems to be an important contribution and could be of theoretical interest.
	On the other hand, a quick lookup on the following handout https://arxiv.org/pdf/2110.11216
	(which in my opinion the authors should cite given that it gives a very good overview to PAC-Bayes bound) suggests that this is not the first time a $\frac{1}{m}$ bound can be achieved.
	Some examples (as per the handout) :

- The noiseless regime where $\mathbb{E}_{\theta\in\rho} [r(\theta)] = 0$ (using Theorem 3.4 as per the handout);

- More generally the Bernstein condition assumption (c.f. Section 4.2 of this handout).

Given this, I suggest that the authors look into these regimes and argue the novelty of their work as compared to these past results.

One other drawback from what I understand of this paper is that, it is not clear to me when does the ``flatness'' assumption hold?
	In addition, for the additional assumption needed for $\ell$ in Gibbs prior, are these assumptions generally applicable?
	To address this, I suggest the authors to characterize 1-2 examples (writing as a lemma) where the gradient vanishes,
	as well as distributions that satisfy such Poincare inequality.
	Without such examples it is hard to gauge how applicable are these assumptions (and therefore harder to gauge contribution).

An example (I would imagine) is when neural nets training reaches a local minimum.
	It is probably useful to cite or establish some results on the flatness degree achieved in terms of number of samples and number of steps.

--------
Edit: the questions I raised turned out to have already been addressed by the author in the first draft, so I am going to change my rating.

**Paper Award:**

No

---

> ### Author Response · Authors · 2024-11-22
> **Rebuttal**
>
> We thank you for your review.
>
> Concerning the comparisons you mentioned, we already cited (Alquier, 2024).
>
> - For the noiseless regime, Theorem 3.4 of Alquier (2024) provides a fast rate for bounded losses for a single dataset size $m$. In contrast, our result holds for nonnegative unbounded losses and holds for all $m$ simultaneously. Furthermore, our proof technique allows us to recover a tighter bound for bounded losses (see proof of Theorem 6, first equation): for any $\ell:\mathcal{H}\times\mathcal{Z}\to[0,1]$, for any $P\in\mathcal{M}(\mathcal{H})$, for any $\lambda$ such that $0<\lambda<2$, with probability at least $1-\delta$, for all $m\in\mathbb{N}^*$ and for all $Q\in\mathcal{M}(\mathcal{H})$, we have
> $R_{\mathcal{D}}(Q) \leq \frac{1}{1-\lambda/2} \left(\hat{R}_{\mathcal{S}_m}(Q) + \frac{\lambda}{2}\frac{KL(Q,P) +\log(1/\delta)}{2\lambda m} \right)$.
>
> - Concerning the Bernstein condition, we already performed a comparison with the PAC-Bayes Bernstein bounds in Appendix A.3.
>
> Thus, in the noiseless case, we also recover a fast rate and it holds for all $m$ simultaneously. In summary, PAC-Bayes Bernstein bounds mixed with the Poincaré inequality are linked to the gradient $\nabla_\mathbf{z} \ell(h,\mathbf{z})$ while the QSB is different and is linked to the gradient $\nabla_h \ell(h,\mathbf{z})$ which is, to our knowledge, novel in PAC-Bayes.
>
> Concerning your remark about when the 'flatness assumption' is reached, we already acknowledged in our conclusion 'However, to complete this analysis, the crucial question is to understand how optimisation algorithms successfully reach flat minima in the
> overparametrised setting. This important question is left as future work.'.
> Indeed, our contribution in this work is to understand and quantify how flat minima help generalisation. Understanding how deep neural networks successfully reach a flat minimum is an essential open question that must be addressed to complete this work.

---

> > ### Comment · Reviewer_sSsB · 2024-12-02
> > **Thank you for your clatification.**
> >
> > Thank you for your rebuttal -- apologies for overlooking the relevant parts as I raise the questions during the initial review. I will change my rating shortly.
> >
> > I still think that it is worth to give 1-2 examples where flatness can be achieved. This would emphasize that the flatness assumption can be achieved (and therefore quantify the usefulness of this work).

---

### Official Review · Reviewer_m8Pd · 2024-11-09
**Flat Minima are sufficient for generalization.**

**Rating:** 6
**Confidence:** 4

**Review:**

## Summary
Flat Minima has long been hypothesized as an indicator for good generalization. Recent empirical results (such as sharpness-aware-minimization) observe that learning algorithms that encourage flatness indeed benefit from better generalization properties. This article investigates the formal link between flatness and generalization. The core contribution of this article is a generalization bound via PAC-Bayes that directly involves flatness terms (via the norm of loss gradients). This article identifies qualitative properties of the posterior that enable fast $\mathcal{O}(\frac{1}{m})$ “transitory” fast-rate generalization bounds (Theorem 6) which indicates that the number of samples needed to guarantee a certain $\epsilon(Q,S_m)$ upper bound on the test error can be fast. Here the $\epsilon(Q, S_m)$ depends on certain properties of the posterior, the data distribution and the empirical risk and in general can be bounded away from zero due to the dependence on loss gradients. In this way, the main contribution of this article is a sufficient condition on the learnt posterior (that depends on the data distribution) that then carries the implication : flatness => good approximate generalization. To obtain an overall test error of $\epsilon$ (rather than the approximate generalization as above) one still needs $\mathcal{O}(\frac{1}{\sqrt{m}})$ samples as noted by the authors.

## Strengths
1. Theoretical frameworks that allow the characterization of multiple phases of generalization are insightful and further our understanding beyond uniform convergence.
2. The presentation and the results are intuitive.


## Weakness
1. Some of the theoretical results require heavy assumptions on the data. These assumptions have classical flavor but the resulting bounds still depend on the expectation over the data. It is unclear if a learning algorithm can guarantee minimal expected sharpness w.r.t parameters on the data distribution as opposed to just minimal sharpness w.r.t parameters at the observed data points. While it is indeed true that algorithms such as Sharpness aware minimization can output posteriors with minimal norm of gradients, such a property is usually only on the training data. Indeed, the extension of such flatness phenomena from the training data to the wider data distribution requires further justification just like the generalization of low empirical risk to low population risk! In Nagarajan et. al and Muthukumar et. al. (see list of references below) for example attempt to generalize flatness from training data to flatness in test data in the context of classification by incurring additional terms in the generalization bound (just like Theorem 8 but with lesser assumptions in a specific setting). Hence, the dependence on expected sharpness over the data distribution in Theorem 6 is a weakness. On a related note, Theorem 6 and Corollary 7 are not empirical - i.e. they cannot be computed or estimated in practice as they depend on assumed properties of the data distributions, the hypothesis class or combinations there-of. Still the spirit of the results developed by the authors indicate the broad links between flatness and generalization. The authors note similar implications in the discussion above Theorem 8 which attempt to alleviate this drawback using additional assumptions on the sensitivity of loss gradients.
2. Much of the result depends on the KL divergence which requires absolute continuity between Q and P. Theorem 14 circumvents this requirement by employing Wasserstein divergence instead. However the assumptions in Theorem 14 appear quite strong - a favorable conditioning of the prior and the data distribution (via $sigma^2$) is assumed and further the generalization gap is assumed to be G-gradient Lipschitz. It is unclear to this reviewer whether such an assumption is reasonable, I urge the authors to supplement their discussion with an example if possible. Further, the bound on the generalization gap depends on the expected norm of the gradient of the generalization gap!  The impact of such a result is unclear. For e.g, is assuming that the generalization gap is Lipschitz significantly stronger than assuming that their gradients are Lipschitz? If not, then I can assume that the generalization gap is itself Lipshcitz and use any predictor $h’$ for which $R_D(h’)$ is known to directly obtain a bound on risk $R_D(h)$ since $|R_D(h)-R_S(h)| \leq |R_D(h') - R_S(h') + L|h-h’|$. Please let me know if I have missed a trivial complication that validates the context of Theorem 14.
3. At several points authors make claims on novelty regarding the usage of gradient norms in generalization theory. A direct quote “We highlight that the gradient term in Theorem 6 is derived with respect to a predictor $h\in \mathcal{H}$ and not $z\in \mathcal{Z}$ which is, to our knowledge, novel in PAC-Bayes”
In my humble opinion, these claims are a slight over-representation of the contribution. In particular, there is a long-standing tradition of developing bounds on the generalization error based on sensitivity measurements i.e., Lipschitz constants all of which are themselves bounds on the norms of the gradient w.r.t parameters across the data domain. In particular, I note that Nagarjan et.al (2019), Banerjee, et.al. (2020) and Muthukumar et. al. (2023) all satisfy the following constraint (1) Use PAC-Bayes as the proof-vehicle, and (2) Show bounds that rely on gradient norms or Lipschitz constants. In each reference, flatness (or sharpness) arises when one derandomizes the PAC-bayes bound from the posterior to a bound holding on the mean predictor $mu_Q$. Such a bound naturally relies on (a) the concentration of the posterior $Q$ around the mean $\mu_Q$, and (2) the sensitivity of loss $\ell$ upon perturbations to the mean predictor. In my opinion these results are comparable to the developed results and this article can benefit from contextualizing them. Banerjee et. al. (2020) in particular appear to link generalization with the norm of the loss Hessian w.r.t parameters. Additionally Bartlett et. al. (2017) and Wei et. al. (2019) bound genearlization error using Lipshcitz constants albeit via a different popular proof vehicle - Rademacher complexity. Wei. et. al. (2019) in particular account for the Jacobian norms of the predictors for general computational graphs.  All the above references utilize sensitivity w.r.t perturbations in the weights directly or indirectly and thus the claim of novelty of using gradients w.r.t. parameters should be amended.
4. The authors claim that their analysis focuses on loss gradients w.r.t parameters rather than loss gradients $\nabla_z \ell(h,z)$ w.r.t data as $\nabla_z \ell(h,z)$ cannot be optimized since data is fixed. This statement requires more context or nuance. In particular, there is much to be gained from learning predictors that are flat within neighborhoods around the data points, i.e. low norm of gradient w.r.t data. Indeed the field of adversarial robustness seeks such predictors and often predictors that are robust to input perturbations generalize better (subject to additional technical nuances). Thus, I think, this particular statement should perhaps be amended to contextualize this work appropriately and indicate the subtleties of seeking flatness w.r.t perturbations in the data domain vs flatness w.r.t perturbations in the parameter domain.
5. The experimental design is incomplete in my opinion. The authors estimate the expectations over the posterior with a single random sample and the expectation over the data with a mini-batch of 512 examples. I think the bar for experimental evidence should be higher, especially for smaller datasets such as MNIST. For e.g. Ujvary et. al (2023) conduct a Hamilton Monte-Carlo sampling to estimate the same posterior. I would suggest that the authors refine the experimental evidence.





## Technical clarifications
1. Quadratic self-boundedness (Assumption 5) is noted as a relaxation of boundedness, can the authors supplement the article with an example where QSB materially improves the analysis, i.e. either (1) A task where using bounds on the loss function or the expected-variance assumption results in a worse bound? Or (2) A task where QSB is necessary because boundedness/expected variances is too strong a requirement.
2. “Theorem 8 not only introduces the first PAC-Bayesian bound involving gradient terms, but it can be transformed into a generalisation metric” , what is a generalization metric and how is it different to just a bound on the generalization error?
3. “Note in this case that, while the KL divergence has an explicit formulation, it requires calculating the exponential moment which is costly in practice. On the contrary, we only need to estimate a second-order moment over the Gibbs posterior” It is unclear why the latter is easier, could the authors expand on this?
4. As a reader, I struggled to see the benefit result (2) in Theorem 11 provides over the basic result of (1) which as I understand is the PAC-Bayes bound for the Gibbs Posterior + bound KL divergence of Gibbs Posterior from Lemma 10. In particular (1) already involves the gradient of the empirical risk w.r.t parameters, what is gained from the additional assumptions in (2)?




## References
1. Peter L Bartlett, Dylan J Foster, and Matus J Telgarsky. Spectrally-normalized margin bounds for neural networks. (NeurIPS, 2017)
2. Vaishnavh Nagarajan and Zico Kolter. Deterministic PAC-bayesian generalization bounds for deep networks via generalizing noise-resilience. (ICLR 2019)
3. Colin Wei and Tengyu Ma. Data-dependent sample complexity of deep neural networks via lipschitz augmentation. (NeurIPS, 2019)
4. Arindam Banerjee, Tiancong Chen, and Yingxue Zhou. De-randomized PAC-Bayes Margin Bounds: Applications to Non-convex and Non-smooth Predictors. (Arxiv 2020)
5. Ramchandran Muthukumar, and Jeremias Sulam.. Sparsity-aware generalization theory for deep neural networks. (COLT 2023).
6. Szilvia Ujváry, Gergely Flamich, Vincent Fortuin, José Miguel Hernández Lobato Estimating optimal PAC-Bayes bounds with Hamiltonian Monte Carlo. (NeurIPS 2023)

**Paper Award:**

No

---

> ### Author Response · Authors · 2024-11-22
> **Rebuttal pt 1**
>
> We thank you for your insightful review. We answer your various questions below.
>
> **Weaknesses**
>
> 1. Thank you for the discussion and pointers. we will add further context to our Theorem 8.
>
> 2. The assumption of gradient Lipschitz losses (also called smoothness) is widely used in optimisation. Especially when considering non-convex objectives for SGD, smoothness often plays a role [1, 2, 3]. Furthermore, in the field of variational inference, the learning objective (or part of it) is often assumed to be smooth to obtain convergence guarantees; see [4] and the references within. Concerning your remark about Lipschitz functions, your intuition is good. In fact, Lipschitz functions are the right notions for deriving Wasserstein PAC-Bayes bounds w.r.t. the 1-Wasserstein distance, see [5, 6, 7]. However, note that in order to obtain convergence properties for optimisation procedures, the smoothness assumption (rather than the Lipschitz one) is necessary in many cases to obtain convergence guarantees [8].
>
> 3. We thank you for all those valuable references and discussions. We have been overenthusiastic in our statement. We will suppress this claim from the next version and will compare our results to those you mentioned.
>
> 4. We agree that we might have been overenthusiastic. However, our remark remains of interest in cases where performing adversarial robustness is costly (e.g. complex data such as an audio signal). There is a tradeoff: if we do not have the resources to perform adversarial robustness, which requires exploring a small sphere around data at each optimisation step, then we are forced to make the QSB assumption to rely only on a flat minimum of $\nabla_h \ell(h,\mathbf{z})$. On the contrary, if we can ensure a flat minimum for both $h\rightarrow \ell(h,\mathbf{z}), \mathbf{z}\rightarrow\ell(h, \mathbf{z})$, then we can relax the QSB assumption, as shown in our Corollary 7 and the discussion below.
>
> 5. We reran the experiments and used 10 random samples of the posterior $Q$. It is worth noting that we do not need Monte Carlo sampling, as we can easily sample from Gaussian distributions. We obtained a similar figure, which is expected, as the values are already averaged over 100 iterations.
>
> **Technical clarifications**
>
> 1. Concerning the QSB condition, we give below an example where the QSB is verified for an unbounded learning problem. Assume $\ell$ is $L$-Lipschitz wrt $h$ and nonnegative, that we are in the realisable case, meaning that there exists $h^*$  such that $\forall \mathbf{z}, \ell(h^*,\mathbf{z})= 0$, and that the mean $m\_Q$ and standard deviation $\sigma$ are both bounded by a certain $K$. Then we have by Lipschitzness and Cauchy Schwarz, for any $\mathbf{z}$, because $\ell(h,\mathbf{z})= \ell(h,\mathbf{z})-\ell(h^*,\mathbf{z})$: $\mathbb{E}\_{Q}[\ell(h,\mathbf{z})]^2 \leq \left(\mathbb{E}\_{Q}[\sqrt{L \|h-h^*\|\ell(h,\mathbf{z})}]\right)^2 \leq L\mathbb{E}\_{Q}[\|h-h^*\|] \mathbb{E}_{Q}[\ell(h,\mathbf{z})]$. Finally, note that, by Jensen and a bias variance decomposition : $\mathbb{E}\_{Q}[\|h-h^*\|] \leq \sqrt{\mathbb{E}\_{Q}[\|h-h^*\|^2]} = \sqrt{\sigma\_Q^2 +\|m\_Q - h^*\|^2}\leq \sqrt{K^2 + (K+\|h^*\|)^2}$. This ensures that the QSB condition is verified in this case with constant $C= L\sqrt{K^2 + (K+\|h^*\|)^2}$.
>
> 2. A generalisation metric is an empirical function of the predictor whose increase or decrease is correlated to the increase or decrease of the generalisation ability of the predictor. In the case of PAC-Bayes, the generalisation metric comes from the generalisation bound. It has been shown, e.g. in [9], that PAC-Bayes generalisation metrics correlate well with generalisation.
>
> 3. Our argument is that, if the loss function increases a lot, then the exponential moment $\mathbb{E}\_{h\sim P}\exp(-\gamma \hat{R}\_{\mathcal{S}\_m}(h)$ can be really small and assimilated to zero by a computer. This is a problem as we divide by the log of this number in the KL, thus it may lead to numerical instability. On the contrary, computing a squared gradient could be far more stable.
>
> 4. The set of assumptions on which Theorem 11-(ii) is based does not require the boundedness of the loss function, as the loss $\ell$ is the sum of a bounded function and a convex function.
>
> Again, we would like to thank you for your careful review and are happy to answer any additional questions.

---

> > ### Author Response · Authors · 2024-11-22
> > **Rebuttal pt 2**
> >
> > **References**
> >
> > [1] S. Ghadimi and G. Lan. Stochastic First- and Zeroth-Order
> > Methods for Nonconvex Stochastic Programming. SIAM J. Optim. 2013
> >
> > [2] Rong Ge, Furong Huang, Chi Jin, and Yang Yuan. Escaping
> > From Saddle Points – Online Stochastic Gradient for
> > Tensor Decomposition. COLT. 2015
> >
> > [3] Jason D Lee, Max Simchowitz, Michael I Jordan, and Ben-
> > jamin Recht. Gradient Descent Only Converges to Minimizers. COLT. 2016
> >
> > [4] J. Domke. Provable Smoothness Guarantees for Black-Box Variational Inference. ICML. 2020
> >
> > [5] Ron Amit, Baruch Epstein, Shay Moran, and Ron Meir. Integral Probability Metrics PAC-Bayes Bounds. NeurIPS. 2022
> >
> > [6] Maxime Haddouche, Benjamin Guedj. Wasserstein PAC-Bayes Learning: A Bridge Between Generalisation and Optimisation. CoRR. 2023
> >
> > [7] Paul Viallard, Maxime Haddouche, Umut Simsekli, and Benjamin Guedj. Learning via Wasserstein-Based High Probability Generalisation Bounds. NeurIPS. 2023
> >
> > [8] G. Garrigos, R. M. Gower. Handbook of Convergence Theorems for (Stochastic) Gradient Methods. CoRR. 2023
> >
> > [9] Gintare Dziugaite, Alexandre Drouin, Brady Neal, Nitarshan Rajkumar, Ethan Caballero, Linbo Wang, Ioannis Mitliagkas, and Daniel Roy. In search of robust measures of generalization. NeurIPS. 2020

---

> > ### Comment · Reviewer_m8Pd · 2024-11-26
> > **Response to Authors**
> >
> > Thank you for your response to my review. Most of my questions are addressed in your response. Below I note a few additional remarks.
> >
> > ````
> > The assumption of gradient Lipschitz losses (also called smoothness) is widely used in optimisation. Especially when considering non-convex objectives for SGD, smoothness often plays a role [1, 2, 3]. Furthermore, in the field of variational inference, the learning objective (or part of it) is often assumed to be smooth to obtain convergence guarantees; see [4] and the references within. Concerning your remark about Lipschitz functions, your intuition is good. In fact, Lipschitz functions are the right notions for deriving Wasserstein PAC-Bayes bounds w.r.t. the 1-Wasserstein distance, see [5, 6, 7]. However, note that in order to obtain convergence properties for optimisation procedures, the smoothness assumption (rather than the Lipschitz one) is necessary in many cases to obtain convergence guarantees [8].
> > ````
> > If I understand correctly, the claim here is that assuming smoothness of the final learnt predictor is more cohesive with optimization phase than assuming Lipschitz. However, the bounds in this article are not developed for the outputs of a specific optimization algorithm and optimization algorithms in practice (incl. Sharpness-aware minimization) can encourage flatness with augmentation/smoothing/regularization but do not guarantee flatness. The bound in theorem 14 depends on the term $E_{h\sim Q} \| \nabla_h R_D(h)− \nabla_h \hat{R}_{S_m}(h)\|$.  For what learning tasks and optimization algorithms is such a term bounded? In particular, how does one control the behavior of $\nabla_h R_D(h)$?
> >
> > Without such a discussion, the assumption that trained predictors output by an optimization algorithm will be flat in this sense appears rather strong.
> >
> > ````
> > We agree that we might have been overenthusiastic. However, our remark remains of interest in cases where performing adversarial robustness is costly (e.g. complex data such as an audio signal). There is a tradeoff: if we do not have the resources to perform adversarial robustness, which requires exploring a small sphere around data at each optimization step, then we are forced to make the QSB assumption to rely only on a flat minimum of $\cdot(.)$
> > ````
> > 1. What does it mean to not have resources to perform adversarial robustness?
> > 2. Can you provide a clear example where adversarial robustness is costly (in the context of answer to 1) but the learning task satisfies QSB?

---

> > > ### Author Response · Authors · 2024-11-27
> > > **Thank you for your answer**
> > >
> > > Thank you for your answer. Let us clarify several points below.
> > >
> > > **About smoothness**
> > >
> > > We would like to clarify that smoothness (or gradient-Lipschitz) is a property of the *loss function* and not one of the final predictor. We are not sure how to understand what smoothness means for a single predictor. On the contrary, making assumptions about the loss function is not new in PAC-Bayes; the most classical examples include subgaussian ([1, Th. 4.1], Alquiier's bound) and bounded ones (McAllester's bound). In Theorems 8 and 14, we do not require strong statistical assumptions on the loss like subgaussianity; we only assume finite second moments for $\ell$ and add the gradient-Lipschitz assumption.
> > >
> > > **About Theorem 14**
> > >
> > > Let us precise several links between generalisation and flat minima in this theorem.
> > >
> > > Theorem 14 informs us that having flat minima on both $R\_{\mathcal{S}\_m}(h)$ and $R\_{\mathcal{D}}(h)$ is a sufficient condition to make this term small and thus, allowing the generalisation gap to be small as well.
> > > However, note that it is possible for $\mathbb{E}\_{h\sim Q}[|| \nabla\_h R\_{\mathcal{S}\_m}(h)- \nabla\_h R\_{\mathcal{D}}(h)||]$ to be small as long as $\nabla_h R\_{\mathcal{S}\_m}(h)$ is close to $\nabla\_h R\_{\mathcal{D}}(h)$. For instance, if $Q= \delta_{\mathbf{w}\_{T}}$ is the Dirac distribution on the output of SGD after $T$ iterations ($\mathbf{w}\_T \in\mathbb{R}^d$), then [2] showed that, for smooth non-convex losses, when the intrinsic noise of SGD is subgaussian, we have with high probability:
> > >
> > > $||\nabla\_h R\_{\mathcal{S}\_m}(\mathbf{w}\_T)- \nabla\_h R\_{\mathcal{D}}(\mathbf{w}\_T)||^2 \leq \mathcal{O}\left( \frac{d\sqrt{T}\log(T)}{m}\right)$
> > >
> > > They also extend this result for SGD with sub-Weibull noise at the cost of an additional assumption on the expansion of the gradients.
> > >
> > > **About controlling the gradient of the theoretical risk**
> > >
> > > As you noticed, if we cannot ensure that $\nabla_h R\_{\mathcal{S}\_m}(h)$ is close to $\nabla\_h R\_{\mathcal{D}}(h)$, then we have to control $\mathbb{E}\_{Q}[||\nabla\_h R\_{\mathcal{D}}(h)||]$ directly. A similar problem to the one that arises in Theorem 6 (as you noticed in point 1 of your rebuttal).
> > >
> > > This theoretical term can be directly controlled for $L$-lipschitz losses, as the gradient is bounded by $L$. Then, starting back from Cthe first equation of the proof of Theorem 6, we can prove, by exploiting the boundedness of the gradient, that for any $P$, any $0<\lambda< 2/L$, with probability at least $1-\delta$, for all $m>0$, and $Q$, we have
> > >
> > > $\mathbb{E}\_{Q}[||\nabla\_h R\_{\mathcal{D}}(h)||] \leq \frac{1}{1- \frac{\lambda L}{2}}\mathbb{E}\_{Q}[||\nabla\_h R\_{\mathcal{S}\_m}(h)||] + \frac{KL(Q,P) + \log(1/\delta)}{m\lambda(1- \frac{\lambda L}{2})}$.
> > >
> > > Thus, in this case, we can control the norm of the gradient of the theoretical risk, for Lipschitz unbounded losses. A similar reasoning applies to the expected square gradient term of Theorem 6.

---

> ### Author Response · Authors · 2024-11-27
> **part 2**
>
> **About your concern about adversarial robustness**
>
> We realise our previous argument was unclear, and we do not want to sound as though we are criticising adversarial robustness methods. Our point was that if we want to perform adversarial robustness, we need to solve a maximisation problem $\sup_{||\xi|| \leq \varepsilon} \ell(h,\mathbf{z} + \xi)$ at each optimisation step, which is solved approximately when no closed-form solution is available and thus requires a (potentially costly) additional optimisation procedure. This is what we meant by needing resources (in the sense of computational resources) to perform optimisation.
>
> In order to address your initial concern, we propose to complete our
> previous claims in the paper concerning the interest of $\nabla_z$ notably by saying that this gradient can be controlled in practice. More precisely, we propose to add a sentence after "Corollary 7 states that if $Q$ reaches a flat minimum (meaning $||\nabla_h\ell(h,\mathbf{z})||$ is small) and this minimum is robust to the training dataset (meaning $||\nabla_\mathbf{z}\ell(h,\mathbf{z})||$ is small), then a transitory fast rate is attainable, requiring only an upper bound on $\mathrm{R}_\mathcal{D}(Q)$." and say:
>
> "In order to have the gradient $||\nabla_\mathbf{z}\ell(h,\mathbf{z})||$ small, one may control it. For instance, one may be interested in specific predictors (e.g., 1-Lipschitz neural networks) where this gradient is bounded by a constant, or one may control it in practice, as it is done in Lipschitz neural networks training or in adversarial robustness training [see e.g., 3, 4]."
>
> **References**
>
> [1] Pierre Alquier, James Ridgway, Nicolas Chopin. On the properties of variational approximations of Gibbs posteriors. JMLR. 2016
> [2] Shaojie Li, Yong Liu, High Probability Guarantees for Nonconvex Stochastic Gradient Descent with Heavy Tails, ICML. 2022
> [3] Aleksander Madry, Aleksandar Makelov, Ludwig Schmidt, Dimitris Tsipras, Adrian Vladu. Towards Deep Learning Models Resistant to Adversarial Attacks. ICLR. 2018
> [4] Qiyang Li, Saminul Haque, Cem Anil, James Lucas, Roger Grosse, Jörn-Henrik Jacobsen. Preventing Gradient Attenuation in Lipschitz Constrained Convolutional Networks. NeurIPS. 2019

---

### Author Response · Authors · 2024-11-22
**Thank you for those insightful reviews**

We warmly thank all the reviewers for their insightful comments, for providing numerous pointers
to additional references (added in the revised version), and for pointing out the typos (which are all corrected).
We hope your main concerns are addressed in this rebuttal and are happy to answer any additional
questions you might have before the final assessment of our paper.

---

### Meta-Review · Area_Chair_naaY · 2024-12-09

**Recommendation:** Accept
**Confidence:** 4

**Metareview:**

This paper has received three outstanding reviews making lots of deep technical comments. These initial reviews were not all enthusiastic about the paper, and in fact one reviewer had some very serious concerns. The authors provided very thorough responses to all reviewers, and engaged in a long and fruitful discussion with the most critical reviewer. This commitment and the quality of the answers impressed the entire review team so much that in the end all of them supported acceptance of the paper. I am happy to concur with this recommendation and propose to accept this work for publication at ALT 2025.

That said, it is absolutely necessary that the authors take the reviewers' comments into account when preparing the final version, given the number and depth of changes that were requested by the reviewers. Seeing the authors' strong commitment during the rebuttal phase, we trust that the authors will indeed make these changes.

**Paper Award:**

No